# Constrained Decoding of Diffusion LLMs with Context-Free Grammars

**Niels Mündler, Jasper Dekoninck, Martin Vechev**
Department of Computer Science
ETH Zurich, Switzerland
{niels.muendler,jasper.dekoninck,martin.vechev}@inf.ethz.ch

🌐 https://constrained-diffusion.ai
⭘ https://github.com/eth-sri/constrained-diffusion

## ABSTRACT

Large language models (LLMs) have shown promising performance across diverse domains. Many practical applications of LLMs, such as code completion and structured data extraction, require adherence to syntactic constraints specified by a formal language. Yet, due to their probabilistic nature, LLM output is not guaranteed to adhere to such formal languages. To address this, prior work has proposed constrained decoding to restrict LLM generation to particular formal languages. However, existing works are not applicable to the emerging paradigm of diffusion LLMs, as this requires supporting token generation in arbitrary order instead of the traditional left-to-right order. In this paper, we address this challenge and present the first constrained decoding method for diffusion models, one that can handle formal languages captured by context-free grammars. Our method relies on solving a newly defined additive infilling problem, which asks whether a partial output with holes can be completed to a valid word in the target language. We reduce this problem to deciding whether the intersection of the target language and a particular regular language is empty, and present an efficient decision algorithm for context-free languages. Empirical results on various applications, such as C++ code infilling and structured data extraction in JSON, demonstrate that our method achieves near-perfect syntactic correctness while consistently preserving or improving functional correctness. Importantly, our efficiency optimizations ensure that the computational overhead remains practical.

## 1 INTRODUCTION

Large language models (LLMs) have recently achieved promising performance across a wide range of tasks (OpenAI, 2023; Google DeepMind, 2025). Due to their capabilities in code synthesis, they achieve impressive scores on diverse code benchmarks (Chen et al., 2021; Vero et al., 2025; Jimenez et al., 2024; Jain et al., 2025) and are integrated into developer workflows as programming copilots (GitHub, 2025; Tabnine, 2025). Further, they are used for processing information into machine-readable formats in various domains (Schmidt et al., 2025; Schilling-Wilhelmi et al., 2024; Goel et al., 2023). Despite these successes, LLMs are inherently probabilistic and offer no guarantees about syntactic validity of generated output, providing an inherent limitation for LLM users.

**Constrained decoding** A promising approach that mitigates this limitation is constrained decoding (Poesia et al., 2022; Beurer-Kellner et al., 2024; Ugare et al., 2024; Melcer et al., 2024). This technique leverages the formal grammar of a target language to guide the generation process, ensuring that the output remains within the language's bounds. Constrained decoding leverages parsing and validation of the generated output in lockstep with the incremental generation process, allowing the model to avoid invalid continuations without restarting inference. It has been widely adopted in practice, with commercial providers offering the option to restrict output to JSON or context-free grammars (OpenAI, 2025a; Anthropic, 2025).

**Current limitations of constrained decoding** Constrained decoding is usually applied to context-free grammars (CFGs), which capture the syntax of common programming languages and popular data formats, like C++ and JSON (Knuth, 1965; Cogumbreiro, 2020). In this context, they can only be applied to left-to-right prefix completion, a common LLM generation setting. However,

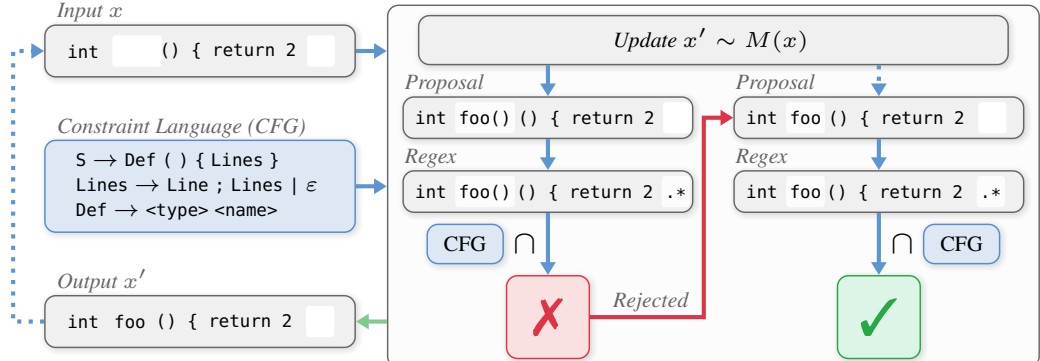

Figure 1: An overview of our approach. In each step, the input consists of a partial text $x$ with arbitrarily many infilling regions and a context-free grammar (CFG) specifying formal constraints. During decoding, we sample an updated input $x'$ from $M$, obtained, e.g., by inserting a token in one of the regions in $x$. Our method then intersects the CFG with the regular language of all possible completions of $x'$. If the intersection is empty, the update is rejected and a new $x'$ is sampled. Otherwise, it is accepted and the decoding continues from $x'$. In the example, the invalid update inserting `"foo()"` is rejected and `"foo"` is accepted instead.

this setting does not capture more advanced use cases with LLMs, such as diffusion LLMs. While Melcer et al. (2024) extend constrained decoding to completions between a fixed prefix and suffix, and Suresh et al. (2025) constrain diffusion LLMs to regular languages, no prior work supports diffusion LLM constraining with CFGs.

**This work: Constrained decoding for MRI and DLMs**  In this work, we present a generalized method for constrained decoding of diffusion LLMs (DLM), which also naturally subsumes the previously unaddressed setting of multi-region infilling (MRI). We first generalize the formal framework of constrained decoding to support unordered updates of a partial output with arbitrarily many infilling regions, capturing both MRI and DLM. The decoding process is illustrated in Figure 1. A model iteratively updates, e.g., inserting a token in a specific location. We verify that the updated output is valid by intersecting the target language's CFG with the language of all possible partial output completions. This intersection is non-empty if and only if a valid completion exists.

A key challenge in this approach is efficiently determining the intersection's emptiness. To this end, we first show that the set of possible completions is described by a regular language, allowing us to describe the intersection using standard formal language operations. We then drastically reduce the worst case cubic cost of checking the emptiness of the resulting intersection language using specialized methods for grammar size reduction and search optimizations, including a custom normal form and an implicit search that avoids generating the entire language.

**Experimentally confirmed consistent improvements**  Our experiments demonstrate a substantial improvement in the reliability of formal language adherence across all evaluated settings. Specifically, the algorithm guarantees valid completions in all settings, up to sampling timeouts. Additionally, it improves functional correctness by up to 7%. Importantly, our approach incurs no initial latency and only modest runtime overhead on tested models, with inference time less than doubling on average, enabling practical usage even for complex constraining grammars.

**Key contributions**  Our three key contributions are: (i) a generalized formal constrained decoding framework for the MRI and DLM settings, (ii) a novel constrained decoding algorithm for these settings, and (iii) an extensive evaluation of our method using state-of-the-art open-weight infilling and diffusion LLMs, demonstrating consistent improvements in syntactic and functional correctness on C++ code generation, JSON schema extraction, and chemical molecule description.

## 2  BACKGROUND

We outline the necessary background relevant to this work, including generation paradigms with LLMs, constrained decoding, and the relevant properties of regular and context-free languages.

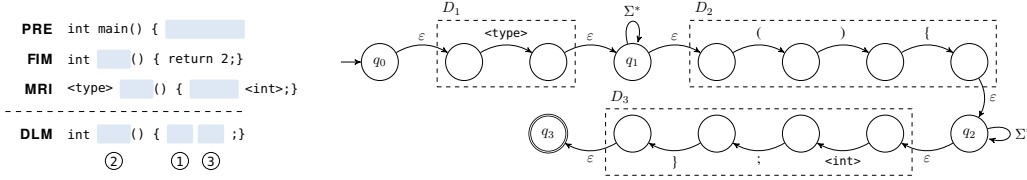

(a) Generation paradigms  (b) NFA accepting all possible completions of the MRI example.

Figure 2: We consider three left-to-right (PRE, FIM, MRI) and one out-of-order (DLM) generation paradigms (a). The NFA in (b) describes the language of all additive completions for the MRI task.

## 2.1 LLM GENERATION PARADIGMS

We focus on four generation settings with LLMs illustrated in Figure 2a. The first three approaches are commonly used with autoregressive models and generate outputs left-to-right.

**PRE, FIM and MRI** The first approach, Prefix generation (PRE) completes a fixed prefix, and is commonly used for synthesizing text or code from scratch. Second, Fill-In-the-Middle (FIM) completes text between a given prefix and suffix, and is widely used in code completion assistants (GitHub, 2025; JetBrains, 2025). Third, Multi-Region Infilling (MRI) generalizes FIM by allowing prefix and suffix constraints as well as fixed segments in between, with the model infilling the gaps. This enables more flexible editing, useful for repository-level code modifications (Wei et al., 2024).

**Generation with DLMs** Diffusion Language Models (DLMs) (Ye et al., 2025; Nie et al., 2025) iteratively insert tokens into an initially empty or partially filled sequence $(x_1, x_2, \ldots, x_n)$ where each $x_i$ is either a token from the vocabulary $V$ or a mask $\perp$. At each step, the model predicts one or more indices $k$ of a masked token, i.e., $x_k = \perp$, and a token $t \in V$ to produce the updated sequence $(x_1, \ldots, x_{k-1}, t, x_{k+1}, \ldots, x_n)$. This process continues until no masks remain. The number of predicted indices and tokens per forward pass is a hyperparameter that controls a trade-off between increased generation speed and quality (Nie et al., 2025). In the example in Figure 2a, the model would generate one index and token at a time, first producing the return keyword ①, then the function name ②, and finally the return value ③.

**Constrained generation** Constrained generation restricts the model to produce outputs that conform to predefined syntactic or structural rules, ensuring syntactically valid code or adherence to structural patterns (Poesia et al., 2022). Formally, the model must generate an output $w \in L$, where $L$ is a formal language defining admissible outputs for the given task. Constrained generation is implemented by restricting the model's probability distribution, either using precomputed masks (Ugare et al., 2024; Poesia et al., 2022), sampling and rejecting invalid tokens (Melcer et al., 2024; Mündler et al., 2025) or a combination of these (Dong et al., 2024; Beurer-Kellner et al., 2024).

## 2.2 REGULAR AND CONTEXT-FREE LANGUAGES

We briefly outline the properties and notation of regular and context-free languages that are relevant to our method. We provide a more detailed introduction in Appendix A.

**Regular Languages** A regular language is a set of strings that can be described by a deterministic finite automaton (DFA). A DFA is defined as a tuple $(Q, \Sigma, \delta, q_0, F)$, where: (1) $Q$ is a finite set of states, (2) $\Sigma$ is a finite alphabet of symbols, (3) $\delta : Q \times \Sigma \to Q$ is a transition function that maps a state and an input symbol to the next state, (4) $q_0 \in Q$ is the initial state, and (5) $F \subseteq Q$ is the set of accepting states. The language of a DFA consists of those strings that transition the automaton from the initial to an accepting state through the transition function. Non-deterministic finite automata (NFA) additionally allow multiple next states for the same state and symbol and traversing $\varepsilon$-transitions without consuming a symbol. An example is depicted in Figure 2b. Every NFA is equivalent to some DFA.

**Context-Free Languages** Context-free languages (CFLs) are a superset of regular languages, including languages that enforce recursive structures, such as balanced parentheses or nested control statements. They can be described by context-free grammars (CFGs). A CFG is a tuple $(V, \Sigma, P, S)$, where: (1) $V$ is a finite set of nonterminals, (2) $\Sigma$ is a finite set of terminals (with $V \cap \Sigma = \varnothing$), (3)

$P$ is a set of productions $A \to \alpha$, with $A \in V$ and $\alpha \in (V \cup \Sigma)^*$, and (4) $S \in V$ is the start symbol. The language is defined as all strings generated by the following procedure: Starting with $S$, apply a rule $A \to \alpha$ from $P$ to replace nonterminal $A$ with $\alpha$, until the result contains only terminals.

## 3 Constrained Decoding for Infilling and Diffusion

In this section, we first define the decision problem that enables MRI and DLM generation settings, and then introduce our algorithm for efficiently deciding the problem. We then provide adapted constrained decoding algorithms for MRI and DLM. Finally, we show how to apply the algorithm to LLMs, where additional challenges arise from the need to handle tokens instead of terminals.

### 3.1 The Constrained Infilling Problem

**Constrained decoding with infilling** First, let us define a partial output $\mathbf{x}$ as a sequence of strings $x_i \in \Sigma^*$ interleaved with infilling regions $\square \notin \Sigma$, i.e., $\mathbf{x} = x_1 \square x_2 \ldots \square x_n$. In constrained decoding, illustrated in Algorithm 1, we complete $\mathbf{x}$ using model $M$ and target language $L$. We iteratively sample an updated partial output $\mathbf{x}'$ from $M$ (Lines 1 and 2). All updated outputs are derived via *additive* modifications to $\mathbf{x}$, meaning they either insert a string into infilling regions, e.g., insert $b$ into $a\square c$ resulting in $a\square b\square c$, or remove a region by merging adjacent strings, e.g., converting $a\square b\square c$ to $a\square bc$.

> **Algorithm 1** Constrained decoding
>
> **Input:** Input $\mathbf{x}$, model $M$, target language $L$
> **Output:** Completed output $\mathbf{x} \in L$
> 1  **while** true **do**
> 2      $\mathbf{x}' \sim M(\mathbf{x})$
> 3      **if** COMPLETABLE($\mathbf{x}', L$) **then**
> 4          $\mathbf{x} \leftarrow \mathbf{x}'$
> 5          **if** $\square \notin \mathbf{x}$ **then**
> 6              **return** $\mathbf{x}$
> 7      **else**
> 8          reject $\mathbf{x}'$

We then check whether the updated output can be completed into a valid word in $L$ (Line 3). If not, we reject the update and remove it from the model distribution, preventing the loop from resampling the update (Line 8). However, if the update is completable, we replace $\mathbf{x}$ with $\mathbf{x}'$ (Line 4) and return the output if the update removes the last infilling region (Lines 5 and 6). This is valid since $\mathbf{x}$ is both completable and has no infilling regions, implying $\mathbf{x} \in L$. Since completability is preserved in updates, there always exists a series of additive updates that completes $\mathbf{x}$ to be in $L$.

**Deciding update validity** To enable constraining additive generation, we need an incremental verifier COMPLETABLE to determine whether the regions in a partial output can be filled to produce a valid output in $L$. We formalize the decision problem solved by COMPLETABLE as follows:

**Definition 1** (Constrained infilling problem). *For a language $L$, partial output $\mathbf{x} = x_1 \square x_2 \ldots \square x_n$ with $x_i \in \Sigma^*$ and $\square$ denoting infilling regions, the constrained infilling problem asks whether there exists a list of $n-1$ words $\mathbf{y} = (y_1, \ldots, y_{n-1})$ such that $w = x_1 \cdot y_1 \cdot x_2 \cdot \ldots \cdot y_{n-1} \cdot x_n$ is in $L$.*

Thus, with the incremental verifier deciding the constrained decision problem, we have effectively reduced constrained decoding with infilling to the constrained infilling problem.

**Applications of the constrained infilling problem** We now reduce constrained decoding for MRI and DLM generation to the constrained infilling problem. For MRI, the list of words corresponds to the list of fixed strings $x_i$, with infilling regions in between. For the DLM setting, we add implicit $\varepsilon$ tokens at the beginning and end of the partially filled sequence and then merge all consecutive non-mask tokens to build $\mathbf{x}$. For example, the sequence $(a, \perp, \perp, b, c, \perp)$ becomes $\mathbf{x} = a\square bc\square\varepsilon$. Note that, similar to prior work (Beurer-Kellner et al., 2024; Ugare et al., 2024), we slightly overapproximate the space of possible completions in these representations by allowing infillings of arbitrary size. In practice, there might be practical limitations to the number of tokens an LLM could insert. We discuss this in more detail in Appendix E.

### 3.2 Deciding the Constrained Infilling Problem Efficiently

**Overview** We now give a brief overview of how to solve the constrained infilling problem efficiently. The problem is determined by two separate constraints: (1) the structural constraints on the output, described by the context-free language $L$, and (2) all possible completions of the partial output $\mathbf{x}$, which form a language $C_{\mathbf{x}}$. For example, $L$ could be the language of syntactically valid C++ programs, and $C_{\mathbf{x}}$ the language of completions of partial program $\mathbf{x} = $int$\square$()$\{\square$2$;\}$. The infilling problem is answered positively if and only if the intersection $L_\cap = L \cap C_{\mathbf{x}}$ is not empty, i.e., some infilling of the partial output exists to generate a valid word in $L$. We will show that $C_{\mathbf{x}}$ is a regular language that we can describe with a simple DFA, and that $L_\cap$ can be described by a

context-free grammar, which we can construct from $L$'s grammar and $C_{\mathbf{x}}$'s DFA. The constrained infilling problem is then reduced to checking whether $L_{\cap}$ is empty, for which we design an efficient algorithm. In the example, a word in the intersection language is `int main() {return 2;}`.

**Constructing the regular language** The language $C_{\mathbf{x}}$ of all possible completions of $\mathbf{x} = x_1 \square \ldots \square x_n$ contains all words that start with $x_1$, end with $x_n$, and contain the strings $x_i$ ($1 \le i \le n$) in the correct order, with arbitrary symbols in between. We prove that $C_{\mathbf{x}}$ is regular by constructing an NFA that accepts $C_{\mathbf{x}}$. We first construct automata $D_i$, which accept exactly $x_i$. Then, we concatenate $D_i$ with an additional state $q_i$ that accepts any string in $\Sigma^*$, i.e., $\delta(q_i, \sigma) = q_i$ for all $\sigma \in \Sigma$. For the concatenation, we add an $\varepsilon$-edge from the accepting states of $D_i$ to $q_i$ and from $q_i$ to the start state of $D_{i+1}$. A visualization for the prior example is shown in Figure 2b. In our algorithm, we construct this NFA for each update. We then transform it into an equivalent DFA and minimize the DFA using standard methods (Hopcroft and Ullman, 1979), as shown in Figure 3b.

**Constructing the intersection language** We leverage the well-established facts that (a) the intersection $L_{\cap}$ of CFL $L$ and regular language $C_{\mathbf{x}}$ is a CFL, whose grammar can be constructed from $L$'s grammar $G$ and $C_{\mathbf{x}}$'s DFA, and (b) that the emptiness of a CFL can be checked in time polynomial to the size of the grammar (Gasarch, 2014; Hopcroft and Ullman, 1979). The symbols in the intersection language have the form ${}^{p}\vec{A}^{\,q}$ for $p, q \in \Sigma$ and $A \in V$, where each symbol intuitively represents deriving a word from $A$ that also traverses the DFA from state $p$ to $q$. The language is nonempty if we can derive a word from ${}^{q_0}\vec{S}^{\,q_f}$ for start symbol $S$ and initial and final state $q_0$ and $q_f$. An example of deriving a word in the intersection language is shown in Figure 3c. The intersection grammar $G_{\cap} = (V_{\cap}, \Sigma, P_{\cap}, S_{\cap})$ will have a cubic size in nonterminals and productions, with $|V_{\cap}| \in O(|V||Q|^2)$ and $|P_{\cap}| \in O(|P||Q|^3 + |P||Q|^2|\Sigma|)$ (Gasarch, 2014; Bar-Hillel et al., 1961). While we can not reduce the worst case complexity of this blowup, we carefully construct the intersection language to keep its size at a minimum, and employ several heuristics to reduce the practical cost of determining its emptiness, explained next.

**Efficient normalization** The standard intersection algorithms require $G$ to be transformed to Chomsky normal form, which only allows rules of the form $A \to BC$ or $A \to a$, where $A, B, C \in V$ and $a \in \Sigma$ (Hopcroft and Ullman, 1979). The resulting grammar may have a quadratic increase in the number of production rules (Lange and Leiß, 2009). To avoid this increase, we extend the standard construction to support CFGs in C2F$^{+\varepsilon}$, a normal form that additionally allows productions of the form $A \to \varepsilon$ and $A \to B$. We provide an example of the normalized C++ grammar in Figure 3a. This normal form can be obtained with only a linear increase in production rules (Lange and Leiß, 2009). Our adaptations to the standard intersection algorithm and a proof of its correctness are provided in Appendix B.1. In Appendix B.2, we describe several further heuristics to reduce the size of the normalized CFG of $G$. After this step, we can intersect the languages and determine the emptiness of the intersection language.

**Avoiding nongenerating nonterminals** The standard algorithm to determine whether language $L$ is empty determines whether the start symbol $S$ is *generating*, i.e., whether there is a sequence of production applications $S \to \cdots \to w$ such that $w \in \Sigma^*$. This property can be decided in time linear to the size of $L$ (Hopcroft and Ullman, 1979). When applied to intersection language $L_{\cap}$, it is important to note that intersection languages contain a significant fraction of non-generating symbols (Nederhof and Satta, 2008; Hanneforth, 2011). We therefore adopt a bottom-up search, that by construction only explores generating symbols (Sipser, 1996), and adapt it to C2F$^{+\varepsilon}$. The algorithm starts with symbols that generate terminals or empty strings directly, i.e., all $A$ with productions $A \to \sigma$ and $A \to \varepsilon$. It marks these symbols as generating and inserts them into a queue. Next, for each symbol $X$ in the queue, it checks whether some production has $X$ on the right-hand side (i.e., either $A \to XC$, $A \to BX$ or $A \to X$), and whether the other symbol ($B$ or $C$) in the production was previously marked as generating. If so, $A$ is marked as generating and added to the queue. As soon as the start symbol $S$ is marked, we conclude that the language is non-empty. We confirm in our experiments that this avoids exploring 98% to 99.99% of productions in $L_{\cap}$.

**Searching through the implicit intersection language** To speed up the emptiness check, we avoid constructing the entire intersection language. Instead, we only construct the parts of the language visited during the search. All symbols in the intersection language have the form ${}^{p}\vec{A}^{\,q}$ for $p, q \in \Sigma$ and $A \in V$. All production rules in the intersection grammar are directly derived from corresponding rules in the original CFG. Specifically, all rules of the form ${}^{p}\vec{A}^{\,q} \to \varepsilon$ and ${}^{p}\vec{A}^{\,q} \to \sigma$

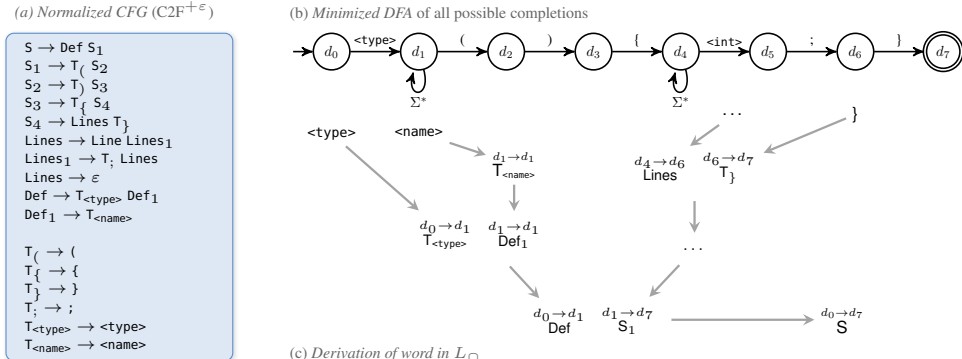

Figure 3: Examples of Figures 1 and 4 processed during our method. (a) The grammar is first normalized into C2F$^{+\varepsilon}$, and (b) the NFA is transformed into a minimal DFA. (c) To determine emptiness of $L_\cap$, the algorithm then searches the initial state $^{d0}\vec{S}^{d7}$ through the productions in reverse, starting from the terminals.

are based on corresponding rules $A \to \varepsilon$ and $A \to \sigma$ in the original grammar without further dependencies, allowing us to iterate over directly generating symbols without constructing the remaining grammar. Further, all other rules are of the form $^p\vec{A}^q \to {}^p\vec{B}^{r\,r}\vec{C}^q$ and $^p\vec{A}^q \to {}^p\vec{B}^q$ based on original productions $A \to BC$ and $A \to B$, for all $p, q, r \in Q$. This enables enumerating all such rules for a given $^p\vec{B}^r$, $^r\vec{C}^q$ or $^p\vec{B}^q$ during the search. We present the corresponding pseudo-code and additional explanations in Appendix B.3.

**Sampling a valid completion from the intersection language**   The algorithm presented above decides intersection emptiness. We now extend it to return a valid completion from the intersection language. To achieve this, we modify the algorithm to track production rules that were applied when marking symbols as generating. These rules describe a parse tree for some word $w$ in the intersection language. We traverse the terminals at the leaf nodes of this tree from left to right to reconstruct a valid completion in the intersection language. This completion is used after a fixed number of rejected updates from the LLM. Since the algorithm leverages the results from the prior emptiness search, it can be run at no additional cost.

### 3.3   APPLICATION OF CONSTRAINED INFILLING TO LLMS

We now briefly outline how to apply the algorithm from §3.2 to LLMs, which generate arbitrary Unicode text rather than language terminals. Full details are provided in Appendix C.

**Lexing**   For typical applications of CFGs, a string of Unicode characters $u$ is converted to terminals $x = t_1 \dots t_k$ in a process called *lexing*. First, note that every terminal $t$ corresponds to a regular language $R_t$ over Unicode characters. During lexing, $t_1$ is obtained by finding the terminal $t$ such that $R_t$ matches a prefix $p$ of $u$, i.e., $u = p \cdot s$. The lexing process then recurses on $u' = s$ to obtain the remainder of $x$, continuing until the string is empty. In principle, to apply this procedure to a sequence with infilling regions $s_1 \perp s_2 \perp \dots s_k$, we would lex each consecutive string $s_i$ to obtain $x = t_1 \square \dots \square t_n$. For example, int $\perp$ () { $\perp$ 2;} would be lexed to `<type>` $\square$ ( ) { $\square$ `<int>` ; }. However, several caveats to this procedure need to be addressed.

**Handling infilling regions**   First, it does not accurately handle partial terminals that border infilling regions, since LLM tokens are Unicode strings that may not align with terminals. For example, the partial LLM output $\perp$2 could correspond to both `<int>` and `<ident>`. The ambiguity stems from the possibility to fill the gap with, e.g., either the token x or 1, resulting in x2 or 12 respectively.

To address this, we treat the text around an infilling region as possibly belonging to an incomplete terminal. Specifically, in the lexing process, we additionally look for terminals $t$ such that the current output $u$ is a prefix of a word in $R_t$ right before an infilling region or a suffix right after a region. Further, we include terminals $t$ that could span across one or more infilling regions by determining if prefixes and suffixes can be infilled to form a single word in $R_t$, as in the example above. We

can thus generate all terminal sequences consistent with a partial output. If any such sequence can be completed to a valid program, then the partial output itself admits a valid completion. In the example above, our algorithm would yield two possible lexings, both □ `<int>` and □ `<ident>`, and intersect both with the context-free grammar..

**Efficiency optimizations**    The number of possible terminal sequences grows quickly with the number of regions and ambiguities. To improve efficiency, we introduce two optimizations. First, for each $x$, we directly construct a single NFA for all possible terminal sequences. This allows us to apply the intersection algorithm once rather than for each sequence. Second, we reduce ambiguity by preprocessing terminals: whenever the accepted language of terminal $t_<$ is contained within terminal $t_\geq$, we remove the overlap from $t_\geq$ and adapt the CFG to allow $t_<$ wherever $t_\geq$ is allowed.

**Sampling a valid completion**    The sampling method from §3.2 returns a sequence of terminals rather than Unicode characters. To sample a Unicode completion, we first concatenate the regular languages of the terminals in the sampled completion. We then construct a regular language for the current partial LLM output and intersect the two languages. Sampling a random string from this intersection yields a valid completion at the Unicode level. In the given example, we would construct regular language of terminals $R$(`<type>`) $R$(`<name>`) $R$(`(`) $R$(`)`) $R$(`{`) $R$(`<int>`) $R$(`}`) and intersect it with `int .* (){ .* 2;}`, resulting in `int x(){2;}`.

## 3.4    Soundness, Completeness, and Alignment

In this subsection, we briefly analyze desirable properties of our algorithm. We first show that the algorithm is sound and complete with respect to the constrained grammar, and fulfills the minimal invasiveness guarantees introduced by Beurer-Kellner et al. (2024).

**Soundness and Completeness**    Our algorithm is sound, i.e., all generated output is valid according to the formal grammar and lexer. This requires the assumption that the lexer of the target language uses maximal munch for lexing, as is common in many programming languages (Park et al., 2024; Melcer et al., 2024). Moreover, our algorithm is complete, i.e., it allows sampling any token that would result in a correct output. A detailed proof of these properties is provided in Appendix C, together with a more precise description of the employed lexing algorithm.

**Minimally invasive**    Our algorithm is *minimally invasive* (Beurer-Kellner et al., 2024). This means that, if the model $M$ without constraints would generate a valid output $w \in L$, it will also produce the valid output when our constraining algorithm is applied. This follows from the algorithm's completeness; it does not reject any partial variants of a valid output. Therefore, in Algorithm 1 for the case where $M$ suggests only valid partial outputs, each $x'$ would be marked as completable, and the algorithm thus would return the same output as if the check was not present.

## 4    Experimental Evaluation

We evaluate our method across a range of tasks and models, first in the MRI setting, and then in DLM, demonstrating improvements in both syntactic and functional correctness. We provide further experimental details, ablate DLM diffusion steps, and provide a case study in Appendix D.

## 4.1    Experimental Setup

**Metrics**    We compute two main metrics to evaluate the effectiveness of our method. First, we determine the percentage of syntactically correct completions (Syntax), which indicates how many of the obtained completions adhere to the specified grammar. We also measure functional correctness (Functional) by either comparing the sample to a golden solution, or by reporting the percentage of solutions that pass all test cases, pass@1, depending on the dataset. All results are averaged over four independent runs with different seeds. We compute confidence intervals at $95\%$, **boldface** the best method, and underline all methods over which the increase is not significant. The usual size of the confidence interval is $1\%$ to $2\%$.

**Compared methods**    We run unconstrained LLM sampling, reported as Vanilla (*Van.*) and constrained decoding with our method (*Con.*). This includes sampling random completions when generation aborts. As an ablation, we report *Con.$^-$*, where these aborted instances are marked as syntactically and functionally invalid. To decide the cutoff for aborting generation, we run our method

on a development set of C++, SMILES and SMILES tasks. We observe that if a task requires more than $50$ resamples, it is functionally correct in only $0.7\%$ of cases. Thus, we significantly speed up the sampling process without losing performance by aborting after $100$. Further, all tested tasks can be solved within $256$ tokens, which we set as a maximum output size for all methods. On the instruction tuned DLMs, we further compare to the baseline of *Grammar Prompting (G.P.)* (Wang et al., 2023), where the model is provided with the Grammar Rules in its prompt.

## 4.2 FILL-IN-THE-MIDDLE AND MULTI-REGION-INFILLING

**Models**    We compare the performance of five recent open-weight infilling models, including STAR-CODER2 7B (Lozhkov et al., 2024b), CODEGEMMA 7B (Zhao et al., 2024), and the DEEPSEEK CODER Family (Guo et al., 2024), covering 7B parameter models from three distinct model families and model sizes from 1.3B to 33B.

**Tasks and benchmarks**    Infilling is commonly used to complete partial code (Bavarian et al., 2022; Fried et al., 2023). We therefore evaluate our method on the C++ translation of the HumanEval dataset (Zheng et al., 2023; Chen et al., 2021), of 164 diverse basic coding tasks. Similar to Bavarian et al. (2022), we construct an infilling dataset by removing random spans from the human-written reference implementation. We evaluate up to three removed spans, resulting in 1-MRI, being equivalent to FIM, and 2-MRI and 3-MRI, with two and three infilling regions respectively. We design a CFG for the subset of C++ syntax needed to solve the tasks in HumanEval. We report adherence to this CFG as syntactic correctness. Functional correctness is measured by computing the pass@1 score on provided test cases (Brown et al., 2020). In Appendix D.4, we also evaluate our method when removing specific lines of code rather than random spans, observing similar improvements.

**Syntactic correctness**    As shown in Table 1, our method increases syntactic correctness significantly across all models and numbers of infilling regions. Our method (Con.) recovers a syntactically valid completion in on average $95.8\%$ of instances. Remaining errors are due to timeouts. Constrained decoding without completions (Con.$^-$) increases syntactic correctness more for code with multiple regions. This coincides with models struggling more, achieving an absolute increase of $5.2\%$, $22.5\%$, and $31.5\%$ for 1-MRI, 2-MRI, and 3-MRI, respectively. These improvements are consistent across model families and sizes, ranging between $17\%$ and $21\%$ per model.

**Functional correctness**    In the lower half of Table 1, we observe that constraining (Con.) consistently increases functional correctness, on average by $2.8\%$, and even without randomly sampling valid completions (Con.$^-$), the average increase is $2.4\%$. This is expected, as syntactically incorrect completions can not be functionally correct and are effectively prevented by our method.

**Runtime overhead**    We compare the time per token between constrained and vanilla decoding. The median runtime overhead of constrained decoding is $4.2\,\mathrm{ms}$, where the overhead on the small DEEPSEEK CODER 1.3B is higher ($5.8\,\mathrm{ms}$) than on the 7B models ($4.6\,\mathrm{ms}$) and DEEPSEEK CODER 33B ($4.3\,\mathrm{ms}$). Moreover, median overhead increases with more complex infilling, growing from $3.1\,\mathrm{ms}$ on 1-MRI to $7.7\,\mathrm{ms}$ on 3-MRI. Further details on runtime are provided in Appendix D.4.

## 4.3 DIFFUSION LANGUAGE MODELS

**Models**    We evaluate our method on the instruction-tuned versions of four state-of-the-art diffusion language models, LLADA 8B (Nie et al., 2025), DREAM 7B (Ye et al., 2025), DREAMCODER 7B (Xie et al., 2025) and DIFFUCODER 7B (Gong et al., 2025). We run all models with 32 steps on 256 tokens and with a temperature of 0.2.

**Tasks and benchmarks**    As DLMs are generic text generation models with many different applications, we design three distinct and diverse tasks:

C++    Based on the dataset used in §4.2, the model should generate the entire function specified in natural language (Chen et al., 2021; Zheng et al., 2023).

JSON    The model should extract relevant information from natural language input, adhering to a JSON-Schema specification (NousResearch, 2024).

SMILES    The model should write down a chemical molecule described in natural language in the SMILES specification language (Weininger, 1988).

For SMILES and JSON we generate synthetic benchmarks using GEMINI-2.5-PRO (Google DeepMind, 2025) with verification to ensure that the generated samples are correct and solvable, resulting

Table 1: Our method consistently improves the percentage of syntactically and functionally correct infillings for varying numbers of regions in MRI under standard decoding (Van.), constrained decoding (Con.⁻), and completing partially completed outputs (Con.).

| | Model | 1-MRI | | | 2-MRI | | | 3-MRI | | |
| --- | --- | --- | --- | --- | --- | --- | --- | --- | --- | --- |
| | | Van. | Con.⁻ | Con. | Van. | Con.⁻ | Con. | Van. | Con.⁻ | Con. |
| Syntax | STARCODER2 7B | 88.2 | 95.0 | **98.9** | 55.4 | 77.7 | **96.3** | 24.5 | 57.2 | **88.3** |
| | CODEGEMMA 7B | 92.5 | 97.2 | **100.0** | 61.5 | 85.6 | **99.0** | 29.9 | 66.4 | **96.0** |
| | DEEPSEEK C. 1.3B | 86.5 | 91.7 | **98.7** | 51.5 | 72.9 | **93.1** | 22.7 | 47.7 | **83.0** |
| | DEEPSEEK C. 6.7B | 93.9 | 98.3 | **100.0** | 62.0 | 84.0 | **97.3** | 32.9 | 64.9 | **94.6** |
| | DEEPSEEK C. 33B | 93.1 | 97.6 | **100.0** | 66.3 | 86.5 | **97.8** | 36.4 | 67.8 | **93.5** |
| Functional | STARCODER2 7B | 53.8 | 56.1 | **56.3** | 20.5 | 23.7 | **24.2** | 7.5 | 10.3 | **11.0** |
| | CODEGEMMA 7B | 57.1 | **59.6** | **59.6** | 24.8 | 29.0 | **29.2** | 8.7 | 12.6 | **12.8** |
| | DEEPSEEK C. 1.3B | 46.5 | 46.4 | **47.2** | 16.1 | 18.4 | **19.2** | 4.9 | 5.4 | **6.5** |
| | DEEPSEEK C. 6.7B | 64.8 | 67.1 | **67.3** | 29.8 | 32.7 | **33.2** | 11.9 | 13.5 | **13.5** |
| | DEEPSEEK C. 33B | 69.8 | 71.2 | **71.4** | 29.8 | 34.0 | **34.3** | 12.6 | 14.3 | **15.4** |

in 167 and 272 instances respectively. We implement the syntax of each language as a CFG and use it to enforce and evaluate the syntactic correctness of the generated output. For C++, we measure functional correctness using pass@1 as in §4.2. For JSON and SMILES, correctness is evaluated by normalizing and comparing to a golden solution. More details about the dataset generation procedure and the correctness evaluation are provided in Appendix D.3.

**Syntax errors** We observe that our method consistently increases syntactic correctness for all tasks and models, as shown in Table 2, in stark contrast to the mixed impact of the non-constraining baseline (G.P.). Without sampling valid completions (Con.⁻), our method increases the percentage of syntactically correct instances by 16.1%, 14.7%, and 26.0% for C++, JSON, and SMILES, respectively. We observe that many models fail to generate syntactically correct output even under constraints, with, for example, only 19.0% correct C++ generations for DREAMCODER 7B. However, sampling valid completions (Con.) recovers the failed instances, increasing to 99.2%. In JSON, constrained decoding with completion achieves 100% syntactic correctness.

**Functional correctness** As shown in the lower half of Table 2, the consistent positive effect of constraining on functional correctness is also present for DLM, with an average increase in functional correctness without completions (Con.⁻) of 1.9%, and a slight additional boost with completions (Con.) to 2.2%. Notably, DREAM 7B performance on JSON increases by 6.9%. DREAM 7B appears to benefit a lot from being provided the target syntax (G.P.) in C++ and SMILES, where it outperforms our constraining (Con.). In the SMILES setting, where models generally perform very poorly at only 1.5% average correctness, syntactic constraints are not able to improve functional correctness significantly, achieving only a modest average increase of 0.2%.

**Runtime overhead** We compare the runtime to complete samples in constrained decoding with the vanilla setting. The median completion overhead is only 0.1 s. We observe both speed-ups of up to 1 s and slowdowns of up to 7.8 s. Speed-ups occur when the decoding is preemptively aborted. Further details for this experiment are provided in Appendix D.4.

**Comparison to DINGO** DINGO (Suresh et al., 2025) is a recently proposed method for constrained decoding of regular languages in DLMs. In Appendix D.5, we show that while our method is more general, being able to enforce context-free constraints, it achieves the same syntactic and similar functional correctness. Furthermore, our method has similar runtime overhead of less than 0.3 ms on the tested tasks, without requiring preprocessing as opposed to DINGO.

## 5 RELATED WORK

**Large language models** LLMs have gained traction for diverse tasks such as code generation (Jiang et al., 2024) and structured output generation (LangChain Developer Documentation, 2025; OpenAI, 2025b; Anthropic, 2025). While the most common approach trains LLMs for PRE generation, many modern code models also support FIM settings (Guo et al., 2024; Lozhkov et al., 2024a;

Table 2: Constrained decoding (Con.⁻) consistently increases the percentage of syntactically correct completions for DLMs over standard decoding (Van.). Non-constraining baselines like Grammar prompting (G.P.) do not consistently improve syntactic or semantic performance.

| | Model | C++ | | | | JSON | | | | SMILES | | | |
|---|---|---|---|---|---|---|---|---|---|---|---|---|---|
| | | Van. | G.P. | Con.⁻ | Con. | Van. | G.P. | Con.⁻ | Con. | Van. | G.P. | Con.⁻ | Con. |
| Syntax | DREAM 7B | 40.5 | 48.6 | 58.7 | **99.4** | 22.4 | 15.7 | 44.9 | **100.0** | 67.7 | 84.2 | 93.7 | **99.4** |
| | DREAMC. 7B | 11.0 | 9.1 | 19.0 | **99.2** | 73.7 | 73.3 | 86.6 | **100.0** | 73.1 | 82.1 | 94.9 | **100.0** |
| | LLADA 8B | 13.3 | 6.7 | 36.1 | **99.7** | 77.5 | 76.0 | 89.0 | **100.0** | 58.2 | 67.4 | 91.3 | **100.0** |
| | DIFFUC. 7B | 39.2 | 33.1 | 54.7 | **99.7** | 64.5 | 44.9 | 76.3 | **100.0** | 69.3 | 57.3 | 92.2 | **99.1** |
| Funct. | DREAM 7B | 6.6 | **11.6** | 8.8 | 9.5 | 7.4 | 6.6 | 11.4 | **14.3** | 0.6 | **2.7** | 1.1 | 1.1 |
| | DREAMC. 7B | 3.7 | 0.6 | 4.9 | **5.2** | 44.6 | 46.2 | 46.7 | **46.7** | **3.4** | 1.2 | **3.4** | **3.4** |
| | LLADA 8B | 3.8 | 2.0 | 5.0 | **5.3** | 43.1 | **50.2** | 49.5 | 49.5 | 0.7 | 0.6 | **1.0** | **1.0** |
| | DIFFUC. 7B | 12.5 | 7.6 | 13.7 | **14.8** | 34.3 | 21.0 | 38.0 | **38.2** | **1.1** | 0.6 | **1.1** | **1.1** |

Zhao et al., 2024). More recently, diffusion language models have been scaled to billion-parameter sizes and demonstrate promising performance on a variety of tasks (Nie et al., 2025; Gong et al., 2025; Xie et al., 2025). Meanwhile, LLMs are prone to errors during generation. For example, they often make mistakes in niche programming languages (Giagnorio et al., 2025) and fundamentally struggle to model specific types of formal languages (Strobl et al., 2024; Ebrahimi et al., 2020).

**Constrained decoding** Constraining LLM generation to context-free languages has been explored extensively in prior work (Beurer-Kellner et al., 2024; Poesia et al., 2022; Beurer-Kellner et al., 2023; Willard and Louf, 2023). Most prior works apply these techniques to the PRE setting (Poesia et al., 2022; Beurer-Kellner et al., 2024; Ugare et al., 2024; Dong et al., 2024; Sun et al., 2025; Park et al., 2025), with some extensions to FIM and context-sensitive features (Melcer et al., 2024; Mündler et al., 2025). Suresh et al. (2025) constrain DLMs specifically, but only to regular languages. To our knowledge, constrained decoding with CFGs has not yet been applied to the MRI or DLM paradigms. Additionally, unlike prior work that employs masking (Ugare et al., 2024; Poesia et al., 2022; Beurer-Kellner et al., 2023) or masking-rejection hybrids (Dong et al., 2024; Beurer-Kellner et al., 2024), our rejection sampling approach incurs no additional latency before starting language inference, significantly reducing friction of switching to a different CFG.

**Leveraging language intersections** Two similar works leverage the intersection of CFLs and regular languages. First, Fazekas et al. (2024) discuss subsequence matching, which asks whether $w$ is a subsequence of any word in language $L$. This is a special case of our decision problem, with $\mathbf{x} = \varepsilon \square w_1 \square \dots \square w_{|w|} \square \varepsilon$, and can also be solved by using the emptiness check for intersection languages. Their work is not applicable to our setting, as it only handles this special case, does not consider practical performance, and does not consider how to handle lexing.

Second, Nederhof and Satta (2008) use intersections of weighted CFGs and DFAs for parsing natural language words, using the intersection language as a succinct representation of admissible parses of lexeme sequences. To reduce the size of these intersections, they also filter non-generating symbols during the intersection construction.

## 6 CONCLUSION

We presented the first constrained decoding method for diffusion models that is able to handle context-free languages such as C++ and JSON. We showed how to reduce the problem of valid completion to an infilling decision problem solvable using formal language techniques. Our optimized algorithm demonstrates a consistent and significant increase in syntactic and functional correctness on a variety of benchmarks and models, while still ensuring efficiency at inference time.

## REPRODUCIBILITY STATEMENT

We describe our implementation in detail in §4 and Appendix D, including details such as hyperparameters and the used compute hardware. Further, all of our experiments were run with fixed seeds and disabled optimizations that would introduce nondeterminism. To ensure complete reproducibility of our results, we publicly release the code implementation of our method, as well as datasets, models, and code used for the evaluation at `https://github.com/eth-sri/constrained-diffusion`. We also include the content of this released code as an anonymized artifact for the double-blind review.

## ACKNOWLEDGEMENTS

This work has been done as part of the grant SAFEAI (Certified Safe, Fair and Robust Artificial Intelligence). The work has received funding from the Swiss State Secretariat for Education, Research and Innovation (SERI), contract no. MB22.00088.

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

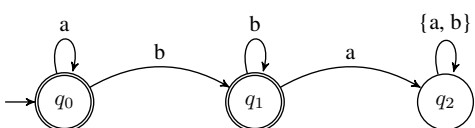

$$V = \{S, B\} \quad \text{(Nonterminals)}$$
$$\Sigma = \{a, b\} \quad \text{(Terminals)}$$
$$S \rightarrow aS \mid bB \mid \varepsilon$$
$$B \rightarrow bB \mid \varepsilon$$

(a) A DFA where $q_0$ is the start state, $\{q_0, q_1, q_2\}$ are the states, and $q_0$ and $q_1$ are the accepting states. The arrows represent the transition function $\delta$.

(b) A CFG with start symbol $S$, terminal alphabet $\Sigma = \{a, b\}$, and nonterminals $V = \{S, B\}$. The production rules are the last two lines.

Figure 4: Two representations of a formal language: a DFA (Figure 4a) and a CFG (Figure 4b). Both accept strings that start with a's and end with b's.

## A EXTENDED BACKGROUND ON FORMAL LANGUAGES

*Formal languages* allow to unambiguously specify valid or invalid strings, usually for ensuring machine-readability, i.e., in the case of JSON schemas, or when specifying the syntactic rules of programming languages. Formal languages are, in their most general form, defined as a set of strings over an alphabet $\Sigma$. For instance, over the alphabet $\Sigma = \{a, b\}$, one can define the formal language $\{\varepsilon, b, aa, bb, aabb, aaaabbb, \ldots\}$ of strings consisting of any number of a's followed by b's. In this section, we provide a short explanation of two key classes of formal languages: regular and context-free languages.

### A.1 REGULAR LANGUAGES

Regular languages are commonly encountered when describing string patterns with regular expressions. For example, the language of a's followed by b's is described by the regular expression a*b*, where the star denotes zero or more repetitions. A regular language can alternatively be described through a Deterministic Finite Automaton (DFA) that accepts the language (Hopcroft and Ullman, 1979). A DFA is a state machine that processes an input string symbol by symbol, transitioning between states based on a deterministic transition function. Thus, a string gets processed by the DFA by starting in the initial state and following the transitions associated with the current input symbol until the end of the string is reached. A string is accepted if the DFA ends in an accepting state after processing the entire string. Formally, a DFA is defined as a tuple $(Q, \Sigma, \delta, q_0, F)$, where: (1) $Q$ is the finite set of states, (2) $\Sigma$ is the finite alphabet of symbols, (3) $\delta : Q \times \Sigma \rightarrow Q$ is the transition function that maps a state and an input symbol to the next state, (4) $q_0 \in Q$ is the initial state, and (5) $F \subseteq Q$ is the set of accepting states. Figure 4a depicts the DFA recognizing the previously introduced language of strings with arbitrarily many a's followed by b's. Note that in this example, the transition function $\delta$ is defined for every state and symbol combination. Per convention, omitted transitions implicitly transfer to a state like $q_2$, from which no accepting state can be reached.

In DFAs, the next transition is thus uniquely determined for each combination of state and input symbol. In contrast, nondeterministic finite automata (NFAs) allow multiple transitions for a state and input symbol combination, making it nondeterministic. One often additionally adds the option to transition between states without consuming any input symbols, through so-called $\varepsilon$-transitions. This added flexibility allows for a more concise depiction and simplifies construction, which is why we use them throughout this work. NFAs accept a word if *any* possible transition according to the input symbols leads to an accepting state. Every NFA (including $\varepsilon$-transitions) can be converted to an equivalent DFA using a standard algorithm (Hopcroft and Ullman, 1979). The NFAs constructed for partial LLM outputs in our method are usually converted into a DFA of around the same number of states, even though the worst-case equivalent DFA can have exponentially many states.

### A.2 CONTEXT FREE LANGUAGES

Context-Free Languages (CFLs) extend regular languages by enabling the expression of recursively nested structures, such as balanced parentheses or properly nested control statements in code. They are described using Context-Free Grammars (CFGs), which consist of production rules that specify how strings in the language can be generated (Hopcroft and Ullman, 1979). For most programming languages, the syntactic rules of the language can be adequately captured by a CFG.

CFGs operate with two types of symbols: terminals, which are the actual characters of the language, and nonterminals, which are used to define the language patterns. A CFG is a formal grammar that consists of a finite set of production rules that describe how strings in the language can be generated. Formally, a CFG is a tuple $(V, \Sigma, P, S)$, where: (1) $V$ is a finite set of nonterminal symbols, (2) $\Sigma$ is a finite set of terminal symbols (with $V \cap \Sigma = \varnothing$), (3) $P$ is a set of production rules of the form $A \rightarrow \alpha$, with $A \in V$ and $\alpha \in (V \cup \Sigma)^*$, and (4) $S \in V$ is the start symbol. To generate a string, one starts with $S$ and applies rules from $P$ until the resulting string contains only terminal symbols. This process defines all valid strings in the language. Figure 4b shows a CFG that generates strings over $\{a, b\}$ starting with arbitrarily many a's followed by b's, demonstrating that the same language recognized by a DFA can also be described by a CFG. To generate the string aabb, one could apply the following sequence of production rules: $S \rightarrow aS \rightarrow aaS \rightarrow aabB \rightarrow aabbB \rightarrow aabb$.

CFGs are often specified in normal forms, which restrict the grammar to certain types of production rules. The benefit of the resulting language is that it reduces edge cases to handle in productions and simplifies proofs about properties of the language. The most common normal form is the Chomsky normal form, where each production rule is of the form $A \rightarrow BC$ or $A \rightarrow a$, with $A, B, C \in V$ and $a \in \Sigma$. Many other normal forms exist, such as C2F, which is based on the Chomsky normal form but additionally allows so-called unit production rules of the form $A \rightarrow B$ (Lange and Leiß, 2009). Languages in Chomsky normal form and C2F can not produce the empty word, as they lack productions that generate the empty string $\varepsilon$ (Hopcroft and Ullman, 1979). We therefore introduce C2F$^{+\varepsilon}$, which additionally allows production rules of the form $A \rightarrow \varepsilon$.

## B    DETAILS ON EFFICIENT INTERSECTION LANGUAGE SEARCHES

In this section, we first detail generic optimizations to reduce the size of context-free grammars, then provide a detailed proof of the correctness of our intersection language construction, and finally provide some more details on the search algorithm we employ to decide emptiness.

### B.1    CONSTRUCTION OF THE INTERSECTION LANGUAGE FOR CFGS IN C2F$^{+\varepsilon}$

We now provide the full constructive proof that the intersection of a CFL and a regular language is a CFL, since it is rarely written out in the literature. We have further adapted it for grammars in C2F$^{+\varepsilon}$. It forms the basis of Algorithm 2, the core algorithm of our method.

**Lemma 1.** *The intersection language $L \cap R$ between a context-free language $L$ and the regular language $R$ is context-free.*

*Proof.* We give a constructive proof by explicitly building a CFG that generates $L \cap R$. We provide the details omitted in the proof given by Gasarch (2014) and extend it to allow grammars in C2F$^{+\varepsilon}$.

Let $L_{\text{CFL}}$ be generated by a CFG $G = (V, \Sigma, P, S)$, and let $L_R$ be accepted by a DFA $(Q, \Sigma, \delta, q_0, F)$. We first convert $G$ to C2F$^{+\varepsilon}$. Then, we construct a new CFG $G_\cap$ whose language is exactly $L_\cap = L_{\text{CFL}} \cap L_R$.

The idea is to simulate the CFG $G$ and DFA $(Q, \Sigma, \delta, q_0, F)$ in parallel. Specifically, we define the nonterminals of $G_\cap$ to be of the form ${}^p\vec{A}{}^q$, where $A \in V$ is a nonterminal of $G$, and $p, q \in Q$ are states of the DFA. We then create production rules in such a way that if there exists a sequence of productions such that ${}^p\vec{A}{}^q \rightarrow \cdots \rightarrow w$, then there exists a sequence of productions in $G$ such that $A \rightarrow \cdots \rightarrow w$ and $w$ takes the DFA from state $p$ to state $q$. We then add a start symbol $S_\cap$ and productions $S_\cap \rightarrow {}^{q_0}\vec{S}{}^f$ for all $f \in F$ to ensure that $L_\cap$ contains exactly the words that can be derived from the start symbol $S$ of $G$ and that also take the DFA from the start state $q_0$ to an accepting state $f \in F$, i.e., all words that are generated by the grammar and all words that are accepted by the DFA.

The productions of $G_\cap$ are defined as follows (adapting (Gasarch, 2014), additional rules in green):

1. For each production $A \rightarrow \sigma$, for all $p, q \in Q$ where $\delta(p, \sigma) = q$, we add ${}^p\vec{A}{}^q \rightarrow \sigma$

2. For each production $A \rightarrow \varepsilon$, for all $p \in Q$, add ${}^p\vec{A}{}^p \rightarrow \varepsilon$

3. For each production $A \to BC$, and for all $p, q, r \in Q$, we add ${}^{p}\vec{A}^{\,r} \to {}^{p}\vec{B}^{\,q}{}^{q}\vec{C}^{\,r}$

4. For each production $A \to B$, for all $p, q \in Q$, add ${}^{p}\vec{A}^{\,q} \to {}^{p}\vec{B}^{\,q}$

The intuition behind the additional rules is that if the automaton is in some state $q$, we can "switch the current symbol" ($A \to B$) or "produce an empty string" ($A \to \varepsilon$) without affecting the state. These productions cover the two additional allowed productions in C2F$^{+\varepsilon}$ grammars, which are not present in CNF grammars.

Finally, we add a new start symbol $S'$ with productions $S' \to {}^{q_0}\vec{S}^{\,f}$ for all $f \in F$.

We show that the language generated by the constructed CFG $L_\cap$ is equivalent to the intersection language of the CFL $L_{\text{CFL}}$ and regular language $L_R$, i.e., $L_\cap = L_{\text{CFL}} \cap L_R$. To do so, we first need some additional notations:

- For any $p, q \in Q$ and $A \in V$, $L({}^{p}\vec{A}^{\,q})$ denotes the language generated by the nonterminal ${}^{p}\vec{A}^{\,q}$ in the constructed CFG, i.e., the set of all words that can be derived with ${}^{p}\vec{A}^{\,q}$ as the start symbol. Note that $L_\cap = \bigcup_{f \in F} L({}^{q_0}\vec{S}^{\,f})$.

- For any $A \in V$, $L(A)$ denotes the language generated by the nonterminal $A$ in the original CFG, i.e., the set of all words that can be derived with $A$ as the start symbol. Note that $L_{\text{CFL}} = L(S)$.

- For any $p, q \in Q$, $L(p \to q)$ denotes the language accepted by the DFA with start state $p$ and final state $q$, i.e., the set of all words that can be accepted by the DFA starting in state $p$ and ending in state $q$. Note that $L_R = \bigcup_{f \in F} L(q_0 \to f)$.

We will show that for any $p, q \in Q$ and $A \in V$

$$L({}^{p}\vec{A}^{\,q}) = L(A) \cap L(p \to q).$$

This immediately implies that $L_\cap = L_{\text{CFL}} \cap L_R$, as

$$L_\cap = \bigcup_{f \in F} L({}^{q_0}\vec{S}^{\,f}) = \bigcup_{f \in F} (L(S) \cap L(q_0 \to f)) = L(S) \cap \bigcup_{f \in F} L(q_0 \to f) = L_{\text{CFL}} \cap L_R.$$

We prove both inclusions separately.

($\subseteq$) We show that for any $p, q \in Q$ and $A \in V$, $L({}^{p}\vec{A}^{\,q}) \subseteq L(A) \cap L(p \to q)$. Let the generation path of $w \in L({}^{p}\vec{A}^{\,q})$ be defined as the sequence of productions that were used to derive $w$ from ${}^{p}\vec{A}^{\,q}$. Denote

$$L_n({}^{p}\vec{A}^{\,q}) = \{w \in L({}^{p}\vec{A}^{\,q}) \mid \text{the generation path of } w \text{ has length at most } n\}.$$

We show the inclusion by induction over the length of the generation path.

**n = 1.** We show that $L_1({}^{p}\vec{A}^{\,q}) \subseteq L(A) \cap L(p \to q)$. Since $w$ is a word, the only possible productions that can be used to derive $w$ from ${}^{p}\vec{A}^{\,q}$ are either rule 1 or rule 2.
In the first case, we know $w = \sigma$, $A \to \sigma$ is a production of the original CFG $G$, and $\delta(p, \sigma) = q$. Hence, $w \in L(A)$ and $w \in L(p \to q)$. In the second case, we have $w = \varepsilon$, $A \to \varepsilon$ is a production of $G$, and $p = q$. Hence, $w \in L(A)$ and $w \in L(p \to q)$.

**n > 1.** Suppose that for all $p, q \in Q$ and $A \in V$, $L_{n-1}({}^{p}\vec{A}^{\,q}) \subseteq L(A) \cap L(p \to q)$. Let $w \in L_n({}^{p}\vec{A}^{\,q})$ be a word with a generation path of length $n > 1$. Then the first production rule applied to $w$ cannot be rules 1 and 2, as these would yield a generation path of length one. Hence, the first rule applied must be either of the rules 3 and 4.
In the former case, we know there exist two words $w_1 \in L_{n-1}({}^{p}\vec{B}^{\,r})$, $w_2 \in L_{n-1}({}^{r}\vec{C}^{\,q})$ such that $w = w_1 \circ w_2$. By induction, we have $w_1 \in L(B) \cap L(p \to r)$ and $w_2 \in L(C) \cap L(r \to q)$. Since $A \to BC$ is a production of $G$, we have $w \in L(A)$

as well. Furthermore, since $w_1$ transitions the DFA from $p$ to $r$ and $w_2$ transitions from $r$ to $q$, we have $w \in L(p \to q)$. Hence, $w \in L(A) \cap L(p \to q)$.

In the latter case, we have $w \in L_{n-1}({}^{p}\vec{B}^{\,q})$ for some nonterminal $B \in V$ such that $A \to B$ is a production of $G$. By the induction hypothesis, we have $w \in L(B) \cap L(p \to q)$. Since $A \to B$ is a production of $G$, we have $w \in L(A)$ as well. Hence, $w \in L(A) \cap L(p \to q)$.

($\supseteq$) We show that $L({}^{p}\vec{A}^{\,q}) \supseteq L(A) \cap L(p \to q)$. Let the generation path of $w$ now be measured with respect to the original CFG, i.e., the sequence of productions that were used to derive $w$ from $A$. Denote

$$L_n(A) = \{w \in L(A) \mid \text{the generation path of } w \text{ has length at most } n\}.$$

We once again show the inclusion by induction over the length of the generation path.

**n = 1.** We show that for any $p, q \in Q$ and $A \in V$, $L_1(A) \cap L(p \to q) \subseteq L({}^{p}\vec{A}^{\,q})$. Since $w$ is a word, the only possible productions that can be used to derive $w$ from $A$ directly are $A \to \sigma$ or $A \to \varepsilon$.

In the former case, we have $w = \sigma$, and since a DFA only consumes symbols one-by-one, there must be a corresponding state transition, i.e., $\delta(p, \sigma) = q$. Hence, $w \in L({}^{p}\vec{A}^{\,q})$ by rule 1.

In the latter case, $w = \varepsilon$, which immediately implies that $p = q$ since a DFA does not contain epsilon transitions. Hence, $w \in L({}^{p}\vec{A}^{\,q})$ by rule 2.

**n > 1.** Suppose that for all $p, q \in Q$ and $A \in V$, $L_{n-1}(A) \cap L(p \to q) \subseteq L({}^{p}\vec{A}^{\,q})$. Let $w \in L_n(A) \cap L(p \to q)$ be a word with a generation path of length $n > 1$. Then the first rule applied cannot be $A \to \sigma$ or $A \to \varepsilon$, as these would yield a generation path of length one. Hence, the first rule applied must be either $A \to BC$ or $A \to B$ for some nonterminals $B, C \in V$.

In the former case, we know there exist two words $w_1 \in L_{n-1}(B), w_2 \in L_{n-1}(C)$ such that $w = w_1 \circ w_2$ and $A \to BC$ is a production in the original CFG. Since $w \in L(p \to q)$, we also know that consuming $w$ transitions the DFA from state $p$ to $q$. We also know that, starting in $q$, after consuming $w_1$, the DFA will arrive at some intermediate state $r$. Clearly therefore $w_1 \in L(p \to r)$. Moreover, since $w = w_1 \circ w_2$ and $w \in L(p \to q)$, also $w_2 \in L(r \to q)$. By induction, we then have $w_1 \in L_{n-1}(B) \cap L(p \to r) \subseteq L({}^{p}\vec{B}^{\,r})$ and similarly $w_2 \in L({}^{r}\vec{C}^{\,q})$. We know that production ${}^{p}\vec{A}^{\,q} \to {}^{p}\vec{B}^{\,r}\,{}^{r}\vec{C}^{\,q}$ is in the intersection language, due to rule 3 quantifying over all states in $Q$. Hence, $w \in L({}^{p}\vec{A}^{\,q})$.

In the latter case, we have $w \in L_{n-1}(B)$ and $w \in L(p \to q)$. By the induction hypothesis, we have $w \in L({}^{p}\vec{B}^{\,q})$. Since $A \to B$ is a production of the original CFG, we have $w \in L({}^{p}\vec{A}^{\,q})$ as well by rule 4.

This completes the proof of the lemma.

$\square$

## B.2 Grammar Size Optimizations

The size of the grammar used for the intersection generation is of high importance to the overall runtime, as the number of productions in the intersection grammar scales cubically with the number of productions in the original grammar. While the size of the intersection grammar also depends on the size of the intersected DFA, generic and efficient methods to minimize DFAs exist. Meanwhile minimization of CFGs is undecidable (Hopcroft and Ullman, 1979).

We therefore apply several heuristics to reduce the grammar size:

- Inlinable terminal elimination: Inline the productions of nonterminals that are only used in a single production. In particular, when $B$ is only used in a single production $A \to \alpha B\beta$, with $B \to \gamma$, remove $B$ and its production and inline it into the production of $A$ to create $A \to \alpha\gamma\beta$.

- Shared 2-gram elimination: For the most frequent $BC$ such that there are several rules of the form $A \to \alpha BC\beta$, (with $\alpha, \beta$ non-empty) introduce $A' \to BC$ and rewrite $A \to \alpha A'\beta$. Repeat until no more such $BC$ with more than one occurrence can be found.

- Left factoring: We eliminate shared prefixes using left factoring (Alfred et al., 2007). Specifically, if two productions of the same nonterminal $A \to \alpha\beta$ and $A \to \alpha\beta'$ share the prefix $\alpha$, we can introduce a new symbol $A'$ and replace the productions to eliminate the duplication, concretely introducing $A \to \alpha A'$ and $A' \to \beta$, $A' \to \beta'$.

After applying these heuristics, we convert the resulting CFG to C2F$^{+\varepsilon}$ using a standard algorithm, consisting of several transformation steps, such as terminal elimination and binarization (Lange and Leiß, 2009). In between each step, we detect and eliminate potentially constructed non-generating symbols.

### B.3 DETAILS ON THE SEARCH ALGORITHM

We explain in detail how the search algorithm for generating nonterminals in the intersection language works. The corresponding pseudo-code is presented in Algorithm 2 and based on the algorithm presented by D.W. (2018). We leverage the construction rules of the intersection language to conduct the search on the *implicit* intersection grammar, i.e., we only build the parts of the grammar that we need to explore. Nonterminals in the intersection language have the form ${}^{p}\vec{A}{}^{q}$ for $p, q \in \Sigma$ and $A \in V$. All production rules of the form ${}^{p}\vec{A}{}^{q} \to \varepsilon$ and ${}^{p}\vec{A}{}^{q} \to \sigma$ are based on the corresponding productions $A \to \sigma$ (Construction 1) and $A \to \varepsilon$ (Construction 2) in the original grammar. We leverage this insight to perform the initialization of the search, which iterates over all production rules of this format, at the beginning of the algorithm in Lines 2–5. Further, all other productions are of the form ${}^{p}\vec{A}{}^{q} \to {}^{p}\vec{B}{}^{qq}\vec{C}{}^{r}$ and ${}^{p}\vec{A}{}^{q} \to {}^{p}\vec{B}{}^{q}$, as constructed by Constructions 3 and 4. Importantly, all rules for all combinations of states $p, q, r$ exist. This allows us to enumerate all such rules for a given symbol $B$ or $C$ on the fly, as done in Lines 9–17, without expending unnecessary execution time. For example, in Line 9, we iterate over all production rules in which the nonterminal ${}^{y}\vec{X}{}^{z}$ occurs. The two states of the DFAs already fixate two of the three states quantified over in Construction 3. Hence, given a production $A \to XC$ in the original grammar, which uses the nonterminal $X$ and additional nonterminal $C$, we need to iterate over a single additional state variable $q$ to evaluate all corresponding constructed productions ${}^{y}\vec{A}{}^{q} \to {}^{y}\vec{X}{}^{zz}\vec{C}{}^{q}$.

## C LEXING WITH LLM TOKENS

The approach described in §3.2 operates directly on the formal language alphabet $\Sigma$. LLMs produce Unicode text that can be misaligned with $\Sigma$. In this section, we describe in more detail how to handle the resulting discrepancies.

### C.1 LEXEMES AND LLM TOKENS

**Discrepancies between alphabet and LLM tokens**  For practical purposes, the alphabet $\Sigma$ of the formal language usually consists of *lexemes*. These represent language components abstractly, i.e., for programming languages, they could be identifiers, literals, operators, and other syntactic elements of the language, such as `if` and `else`. Before parsing a Unicode string, it thus first needs to be converted into a string of lexemes. This process is called *lexing*.

The code generation paradigms MRI and DLM generate code on a Unicode level and thus require lexing before our method can be applied. In addition to the normal lexing process, our approach needs to handle the partial nature of the LLM outputs, taking into account potential partial lexemes and consequently several possible lexing sequences for the same character-level output. In the remainder of this section, we first explain how to convert the partial LLM output to a set of possible lexeme sequences, and then how to apply the constrained infilling algorithm to these lexeme sequences.

**Lexemes and lexing**  Each lexeme is associated with a regular language $R$ where $\Sigma_R$ is the set of Unicode characters. For example, the <number> lexeme is associated with regular expression \d+,

and the `<identifier>` lexeme with `[a-zA-Z_]\w*`. Lexing is the process of converting a Unicode-level string into a sequence of lexemes, i.e., a sequence of strings that match the regular expressions of the lexemes. We call such a sequence of lexemes a *lexeme sequence*. The lexing algorithm extracts these sequences by iteratively matching the maximum match for all lexemes that match a nonempty string at the beginning of the currently remaining output. Whitespace between lexemes is commonly stripped. For example, the character-level string `"1234 hello12"` would be lexed into the lexeme sequence (`<number>`,`<identifier>`).

## C.2 Converting Partial Outputs to Lexemes and DFAs

**Lexing partial outputs** For a partial output x with infilling regions, we extract the represented lexeme sequences for each chunk of continuous text. For instance, the output `"x = 1234□hello12"` would be split into the chunks `"x = 1234"` and `"hello12"`, which would be lexed into the two lexeme sequences (`<identifier>`, `<=>`, `<number>`) and (`<identifier>`). Note that the resulting list of lexeme sequences is a list of words in $\Sigma$ that can be directly used to construct the regular language for the infilling problem as described in §3.2, for example here forming the infilling problem `<identifier><=><number>□<identifier>`.

**Handling lexemes spanning infilling regions** However, infilling regions complicate the lexing process. Concretely, we need to handle strings that match lexemes partially on the border of infilling regions.

Concretely, strings before an infilling region may end with a string that matches a prefix of some lexeme. For example the output `"□123"` could be lexed as (`<number>`). However, the region could be filled with token `"a"`, resulting in the overall lexing (`<identifier>`). Similarly, strings may match suffixes of lexemes after infilling regions.

Additionally, lexemes may span over an entire infilling region. For example, for the output `"123□789"`, a trivial possible lexing is (`<number>`, `<number>`). However, it is also possible to insert a token `"456"` into the region, such that the lexing of the final character-level text is just a single lexeme sequence (`<number>`). This also holds for any number of infilled gaps, e.g., `"123□4□5□6□789"`. In particular, for any chunk $\alpha\beta$ ending with a prefix $\beta$ of a lexeme `<a>`, consecutive chunks $\gamma_i$ that are prefix of a suffix of `<a>` and a final chunk $\eta\zeta$ starting with a suffix $\eta$ of `<a>`, then a valid corresponding lexeme sequence for the entire chunk sequence could be $(\text{lex}(\alpha), \texttt{<a>}, \text{lex}(\zeta))$.

We also need to ensure the prefixes and suffixes of the lexeme are compatible. For instance, for fixed-width lexemes such as `<while>`, we can not insert a token into the sequence `"whil□hile"` to obtain a sequence with only a single lexeme, even though both `"whil"` and `"hile"` are true prefixes and suffixes of the lexeme `while`. We resolve this by determining the intersection of the concrete partially generated output with the lexeme's regular language. Concretely, we construct partial character-level output $\beta□\gamma_1□\ldots□\gamma_n□\eta$ and compute the intersection with the regular language of `<a>` using standard algorithms for the intersection and emptiness of regular languages (Hopcroft and Ullman, 1979). If the language is not empty, (`<a>`) is a valid lexing of the entire sequence. This effectively generalizes a similar solution to the one proposed by Melcer et al. (2024), in which they explicitly store the reached states within each prefix and suffix and ensure their reachability.

**Lexing algorithm** We use some helper operations on character-level DFAs for the lexing algorithm. For DFA $D$, we define the function MATCH, which returns $l$, the number of characters in the string that the suffix language automaton matches maximally. For example, `\d+`.MATCH(123) $= 3$ and `\d+`.MATCH(1hello) $= 1$. The function PREFIX($D$) returns the true prefix language of $D$, where a *true* prefix is a prefix that can be completed to a full match by appending at least one more character. For example, `123` is a true prefix for `\d+` but not for `\d\d\d`. `12` is a true prefix for both regular expressions. The function SUFFIX($D$) analogously returns the true suffix language of $D$. Further, we denote as $w_{\leq i}$ the string formed by the first $i$ characters of $w$ and $w_{>i}$ the string formed by all characters after the first $i$ characters in $w$.

The lexing algorithm applied to each chunk of continuous text in x is described in Algorithm 3. The main mode of operation is to keep track in $S$ of possible lexings and remainders to be processed, starting with the empty lexing and the entire string to be processed in Line 1. The method then

---

**Algorithm 2** Deciding intersection emptiness of a CFG and DFA. The CFG is in CNF. $G.\text{ADD}(x)$ inserts $x$ into $G$ and returns true if $x$ was not in $G$ previously.

---

**Input:** CFG $C$, DFA $R = (Q, \Sigma, \delta, q_0, F)$
**Output:** $L(C) \cap L(R) = \varnothing$
1  $G \leftarrow \varnothing$
2  **for** all productions $A \to \sigma$ **do**  $\qquad\qquad\qquad\qquad$ ▷ Mark terminal and epsilon productions.
3  $\quad G \leftarrow G \cup \{{}^{p}\vec{A}^{q} \mid \delta(p, \sigma) = q\}$
4  **for** all productions $A \to \varepsilon$ **do**
5  $\quad G \leftarrow G \cup \{{}^{p}\vec{A}^{p} \mid p \in Q\}$
6  $s \leftarrow G.\text{COPY}()$
7  **while** $s \neq \varnothing$ **do**  $\qquad\qquad\qquad\qquad\qquad$ ▷ Explore all remaining productions
8  $\quad {}^{y}\vec{X}^{z} \leftarrow s.\text{POP}()$
9  $\quad$ **for** all productions $A \to XC$, all $q \in Q$ **do**
10 $\quad\quad$ **if** ${}^{z}\vec{C}^{q} \in G$ **and** $G.\text{ADD}({}^{y}\vec{A}^{q})$ **then**
11 $\quad\quad\quad$ $s.\text{ADD}({}^{y}\vec{A}^{q})$
12 $\quad$ **for** all productions $A \to BX$, all $q \in Q$ **do**
13 $\quad\quad$ **if** ${}^{q}\vec{B}^{y} \in G$ **and** $G.\text{ADD}({}^{q}\vec{A}^{z})$ **then**
14 $\quad\quad\quad$ $s.\text{ADD}({}^{q}\vec{A}^{z})$
15 $\quad$ **for** all productions $A \to X$ **do**
16 $\quad\quad$ **if** $G.\text{ADD}({}^{y}\vec{A}^{z})$ **then**
17 $\quad\quad\quad$ $s.\text{ADD}({}^{y}\vec{A}^{z})$
18 **return** $G \cap \{{}^{q_0}\vec{S}^{f} \mid f \in F\} = \varnothing$  $\quad$ ▷ Whether any start symbol of $L(C) \cap L(R)$ is generating

---

**Algorithm 3** Extracting lexings of a chunk within a partial output.

---

**Input:** Input string $w$, Terminals $T$
**Output:** Set $\{(x_i, s_i, p_i)\}_{0 \leq i \leq n}$ of $n$ possible lexeme sequences $x_i$ and optional partial matches to the first ($s_i$) or last ($p_i$) lexeme
1  $S \leftarrow \{(\varepsilon, w, \text{None}, \text{None})\}$
2  **for** $t \in T$ **do**  $\qquad\qquad\qquad\qquad$ ▷ Determine if the string starts with a suffix of any terminal
3  $\quad$ **if** $\text{SUFFIX}(\text{PREFIX}(t)).\text{MATCH}(w) = |w|$ **then**  $\quad$ ▷ If the suffix prefix spans the entire word.
4  $\quad\quad$ $S.\text{ADD}(t, \varepsilon, w, w)$
5  $\quad$ $l \leftarrow \text{SUFFIX}(t).\text{MATCH}(w)$
6  $\quad$ **if** $l > 0$ **then**  $\qquad\qquad\qquad\qquad\qquad$ ▷ If the suffix matches a non-zero prefix of $w$
7  $\quad\quad$ $S.\text{ADD}(t, w_{>l}, w_{\leq l}, \text{None})$
8  **while** $S \neq \varnothing$ **do**
9  $\quad$ $(x, w, s, p) \leftarrow S.\text{POP}()$
10 $\quad$ **if** $w = \varepsilon$ **then yield** $(x, s, p)$
11 $\quad$ **for** $t \in T$ **do**
12 $\quad\quad$ **if** $\text{PREFIX}(t).\text{MATCH}(w) = |w|$ **then**  $\quad$ ▷ If the prefix spans the entire remaining word.
13 $\quad\quad\quad$ $S.\text{ADD}(x \circ t, \varepsilon, s, w)$
14 $\quad\quad$ $l \leftarrow t.\text{MATCH}(w)$
15 $\quad\quad$ **if** $l > 0$ **then**  $\qquad\qquad\qquad\qquad$ ▷ If the suffix matches a non-zero prefix of $w$
16 $\quad\quad\quad$ $S.\text{ADD}(x \circ t, w_{>l}, s, \text{None})$

---

---

**Algorithm 4** Extracting lexings of an output with infilling regions.

---

**Input:** Input string $w_1 \square w_2 \square \ldots \square w_n$, Terminals $T$
**Output:** A list of partial lexeme sequences $x$ that match the input string.

▷ Step 1: Collect all possible lexings of partial outputs

1  $L \leftarrow \{\}$
2  **for** $i$ in 1 to $n$ **do**
3      $L.\text{ADD}(\text{LEX\_PARTIAL}(w_i, T))$                   ▷ Apply Algorithm 3

▷ Step 2: Determine all possible subsequences of lexings

4  $C \leftarrow \text{CROSS\_PRODUCT}(L)$
5  **for** $[(x_i, s_i, p_i)]_{i=1}^{n}$ in $C$ **do**

    ▷ Step 2.1: Determine all possible merges of subsequences

6      **if** not $p_0 = s_n =$ None **continue**               ▷ Reject unmatched front or back.
7      $M \leftarrow \{\}$
8      **for** i in 1 to $n$ **do**                    ▷ Check for mergeable subsequences.
9         **if** $s_i =$ None **continue**
10        **for** j in $i + 1$ to $n$ **do**
11           **if** $p_j =$ None **break**           ▷ Are the same lexemes matching partially?.
12           **if** $x_i^{(-1)} \neq x_j^{(0)}$ **break**
13           **if** $p_j = w_j$ and $s_i = w_i$           ▷ Case distinction for all scenarios.
14               $o \leftarrow \square s_i \square w_{i+1} \square \ldots \square p_j \square$
15           **else if** $p_j = w_j$ and $s_i \neq w_i$
16               $o \leftarrow \quad s_i \square w_{i+1} \square \ldots \square p_j \square$
17           **else if** $p_j \neq w_j$ and $s_i = w_i$
18               $o \leftarrow \square s_i \square w_{i+1} \square \ldots \square p_j$
19           **else**
20               $o \leftarrow \quad s_i \square w_{i+1} \square \ldots \square p_j$
21           **if** $x_i^{(-1)} \cap o \neq \varnothing$        ▷ If lexeme can match the partial output
22              $M.\text{ADD}(i, j)$              we can merge from $i$ to $j$.
23           **if** $p_j \neq w_j$ **break**        ▷ No further merges possible with $s_i$.

    ▷ Step 2.2: Determine all possible combinations of merges

24      $N \leftarrow \{[\,]\}$                 ▷ No merge is a possible sequence
25      **for** $(i, j)$ in $M$ **do**
26         $N.\text{ADD}([(i, j)])$
27         **for** $[(i'_1, j'_1), \ldots, (i'_k, j'_k)]$ in $N$ **do**
28           **if** $[i, j] \cap ([i'_1, j'_1] \cup \cdots \cup [i'_k, j'_k]) = \varnothing$    ▷ Non-overlapping merges.
29              $N.\text{ADD}([(i'_1, j'_1), \ldots, (i'_k, j'_k), (i, j)])$
30      **for** $[(i_1, j_1), \ldots, (i_k, j_k)]$ in $N$ **do**
31         **yield** $x_1 \square \ldots \square x_{i_1-1} \square x_{i_1} x_{j_1}^{1:} \square x_{j_1+1} \square \ldots \square x_{i_k-1} \square x_{i_k} x_{j_k}^{1:} \square x_{j_k+1} \square \ldots \square x_n$

---

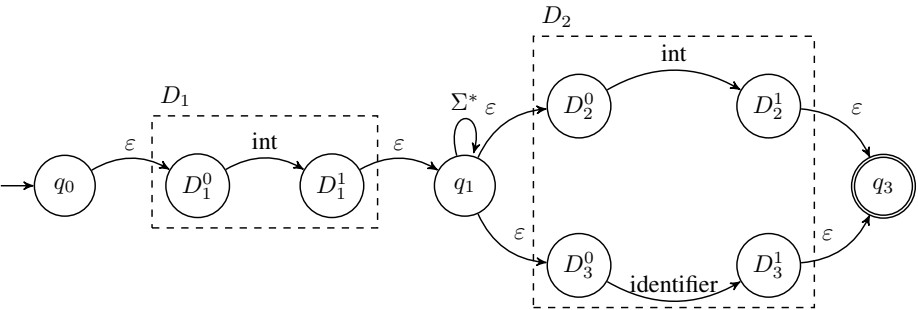

Figure 5: A union automaton in the second half of the DFA for output `"123 □ 789"`, accounts for the possibility to lex the second half as either `<int>` or `<identifier>`. The resulting automaton accepts both valid lexeme sequences `<int><int>` and `<int><identifier>`.

iterates over all these lexings in Line 8, returns them if the remainder is empty (Line 10) or extends them if a non-empty remainder remains (Line 11). Crucially, Lines 2–7 check whether the text starts with the suffix of any lexeme. Additionally, Line 12 checks whether the remainder of the current text is the prefix to some lexeme.

**Applying the constrained infilling algorithm**  Algorithm 4 describes how to apply the lexing algorithm to a partial output with infilling regions. First, we apply the lexing algorithm to each continuous chunk of text in Algorithm 4, resulting in a list of sets of possible lexeme sequences for each chunk. Next, we take the cross product of these sets to obtain all possible combinations of lexeme sequences for the entire output. Each combination consists of a list of lexeme sequences, along with potential partial matches to the first and last lexemes in each chunk. Further steps in the algorithm enable the merging of lexemes that span infilling regions. We first find all possible indices $(i, j)$ for which such merges are possible by checking the compatibility of the partial matches and the regular language of the lexeme, as described previously. We then construct all possible combinations of non-overlapping merges and yield the resulting lexeme-level partial outputs.

**Determining non-emptiness**  The prior algorithm returns a set of possible lexeme-level partial outputs for the given character-level partial output. If any of the resulting sequences results in a intersection language $L_\cap$, then the current character-level partial output is valid, and we can continue generation. If no lexeme sequence results in a non-empty intersection, then we need to reject the current output. Thus, we have to apply the infilling algorithm to each of the word lists derived from the lexing process. In practice, we may derive a large number of lexeme sequences, as different possibilities from text chunks get combined and result in a combinatorial explosion. To further optimize the lexing process, we add two additional optimizations, which we describe in the following paragraphs.

**Optimizing subset lexemes**  We avoid a combinatorial explosion of possible lexeme sequences by automatically removing lexemes where the accepted language is a subset of the accepted language of another lexeme. For example, in SMILES, the string `"5"` could be interpreted as `<digit>` or as `<fifteen>`, which is a special lexeme only allowing numbers from 1 to 15. We resolve this by automatically detecting lexemes $\alpha$ that accept a subset of valid strings of another lexeme $\beta$, and a) remove the subset $\alpha$ from the accepted language of lexeme $\beta$, and b) allow the lexeme $\alpha$ at any position in the grammar where either the subset token or the full token is allowed, in particular we substitute terminal $\beta$ with $\alpha \mid \beta$. This optimization reduces the number of extracted sets of possible lexeme sequences for each continuous chunk of text.

We further manually reduce the number of lexemes that overlap and lexemes that are prefixes or suffixes of other lexemes, such as `<++>` and `<+>`, to further optimize performance.

**Combining lexeme sequences to a single NFA**  To avoid explicitly enumerating all possible combinations of lexeme sequences of a string, we directly derive a single, larger NFA that accepts all possible combinations of lexeme sequences at the same time. This NFA is structurally similar to the NFAs of each lexeme sequence, but adds alternative paths for mergeable lexemes.

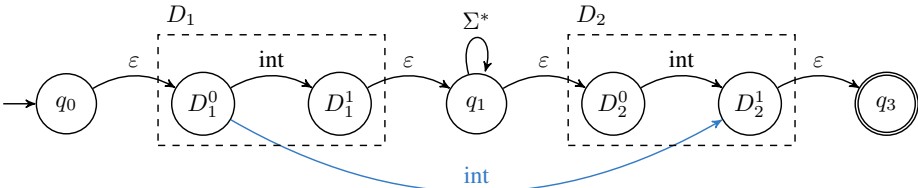

Figure 6: A skip connection, highlighted in blue, in the DFA for output `"123 □ 789"`, accounts for the possibility to lex the input as a single `<int>`. The resulting automaton accepts both valid lexeme sequences for a single int and two ints with intermediate tokens. This construction can be combined with Figure 5.

If a text chunk has two or more admissible lexings, we replace the constructed $D_i$ with an NFA that accepts the union of admissible lexings. For example, the output `"123□789"` must also admit recognizing the second chunk as the suffix of an identifier. Thus, we obtain the two sequence sets `{(<int>)}` and `{(<int>), (<identifier>)}`. By generating a single NFA that accepts both sequences `(<int>)` and `(<identifier>)`, we can construct a single NFA by applying the concatenation construction to the standard NFA for the first lexeme sequence and the unionized NFA for the second sequence. The resulting NFA is presented in Figure 5.

Another example is depicted in Figure 6. Here, for the previously shown output `"123□789"`, the first chunk ends with a prefix of the lexeme `<int>`, and the second chunk starts with a suffix of the same lexeme. In addition to the standard construction for the possible extracted list `(<int>, <int>)`, we add an `<int>`-edge from the second-to-last state of $D_0$ to the second state of $D_1$, resulting in an alternative path that accepts the list `(<int>)`. These paths are constructed by maintaining a list of suffixes of the previous $D_i$ when constructing $D_{i+1}$, and adding the edge if a suffix matches a prefix of the lexing of $D_{i+1}$.

In contrast to the combinatorial explosion observed when considering all possible combinations of consecutive parsed lexeme sequences, this NFA grows only linearly in the number of sequences. We also observe that the generated corresponding DFA has a similar number of states, confirming that this avoids expensive combination enumeration.

### C.3 SOUNDNESS AND COMPLETENESS OF LEXING APPROACH

In this section, we show soundness and completeness of our lexing approach under reasonable assumptions. Throughout, *lex* denotes the lexing function that maps a Unicode string to a lexeme sequence. Each terminal $t$ has a regular language $R_t \subseteq \Sigma^*$ over the Unicode alphabet $\Sigma$. For a lexeme $x$, we will denote $R(x)$ as the regular language of the terminal associated with $x$. Further, for a sequence of lexemes $x_1 x_2 \ldots x_n$, we denote the regular language described by the sequence as $R(x_1 x_2 \ldots x_n) = R(x_1) \circ R(x_2) \circ \ldots \circ R(x_n)$, where $\circ$ denotes concatenation of languages. We denote substrings of $w_j w_{j+1} \ldots w_k$ of Unicode strings $w = w_1 w_2 \ldots w_n, k \leq n$ with $w_{j:k}$, where each $w_i$ is a single unicode character. Finally, we say that a lexeme $x_i$ consumes a substring $w_{j:k}$ if $x_i$ is the part of $x$ that exactly matches the characters $w_{j:k}$.

We make two standard assumptions about the lexer.

**Assumption 1** (Maximum Munch). *The lexer matches lexemes by greedily matching the longest possible lexeme at each position. Concretely, let $w = w_1 w_2 \ldots w_n$ be a Unicode string, where each $w_i$ is a single character, and suppose $\text{lex}(w) = x_1 x_2 \ldots x_m$. Then:*

1. *There exist indices $k_0, k_1, \ldots, k_m$ with $0 = k_0 < k_1 < \ldots < k_m = n$ such that for all $1 \leq i \leq m$, the substring $w_{k_{i-1}+1:k_i} \in R(x_i)$,*

2. *For each $k_i$, there is no $k > k_i$ and terminal $t$ such that the substring $w_{k_{i-1}+1:k} \in R_t$,*

Intuitively, Assumption 1 states that the lexer is greedy: at each position it chooses the terminal that extends the match as far as possible before emitting a lexeme and moving on.

**Assumption 2** (Orthogonal Terminals). *Let $t_1$ and $t_2$ be two terminals with regular languages $R_1$ and $R_2$, and $t_1 \neq t_2$. Then no suffix of any string in $R_1$ is a prefix of any string in $R_2$.*

This assumption can be enforced for any given CFG and set of terminals by preprocessing the terminals: whenever two terminals share a prefix or suffix in this way, we factor out the intersection into a new terminal and adjust the grammar accordingly. These assumptions together imply a useful property:

**Corrolary 1** (Unique Lexing). *Let $w$ be a Unicode string, and $x_1 \ldots x_m$ be a lexeme sequence such that $w \in R(x_1 \ldots x_m)$. Suppose further that, whenever $w_{i:j}$ is consumed by $x_k$ in the sequence, there is no $l > j$ such that $w_{i:l} \in R(x_k)$. Then, $\text{lex}(w) = x_1 \ldots x_m$.*

*Proof.* Let $\text{lex}(w) = y_1 \ldots y_r$. We show that $x_1$ and $y_1$ are equal and consume the same number of characters from $w$. Recursively applying this argument then shows that $r = m$ and $y_i = x_i$ for all $1 \leq i \leq m$.

Let $x_1$ consume $w_{1:k_1}$ and $y_1$ consume $w_{1:k_2}$. Suppose, for the sake of contradiction, that either $k_1 \neq k_2$ or $x_1 \neq y_1$. However, either $w_{1:k_1}$ is a prefix of $w_{1:k_2}$ or vice versa. By Assumption 2, this implies that $x_1 = y_1$. Now, because of Assumption 1 and the assumptions in the corollary, $k_1 = k_2$ must hold as well. □

We now characterize the outputs of Algorithm 3 and Algorithm 4 under these assumptions. We denote the first and last lexeme in a lexeme sequence $x$ with $x^{(1)}$ and $x^{(-1)}$, respectively.

**Lemma 2.** *Given a Unicode string $w$, Algorithm 3 returns* exactly *all tuples $(x, s, p)$ for which there exist strings $u$ and $v$ such that $u \circ w \circ v \in R(x)$ and*

1. *If $w_i \ldots w_j$ is consumed by $x_k$, then there is no $l > j$ such that $w_i \ldots w_l \in R(x_k)$.*

2. *If $s = p = w$, then $|x| = 1$ and $x^{(1)} = x^{(-1)}$ consumes $u \circ w \circ v$.*

3. *If $s = None$, then $u = \varepsilon$. Otherwise, unless $s = p = w$, $x^{(1)}$ consumes $u \circ s$.*

4. *If $p = None$, then $v = \varepsilon$. Otherwise, unless $s = p = w$, $x^{(-1)}$ consumes $p \circ v$.*

*Proof.* We prove both directions: (1) every tuple produced by the algorithm satisfies the stated conditions, and (2) every tuple satisfying the conditions is eventually produced by the algorithm.

**No other tuples** We show that any element $(x, w', s, p) \in S$ (as maintained in Algorithm 3) satisfies the following:

- $w'$ can be written as $w' = w_{j:|w|}$ for some $1 \leq j \leq |w| + 1$, where we set $w_{|w|+1:|w|} = \varepsilon$.

- There exist $u$ and $v$ such that $u \circ w_{1:j-1} \circ v \in R(x)$ and if $j \neq |w| + 1$, then $v = \varepsilon$.

- Condition 2, Condition 3, and Condition 4 hold with respect to $w_{1:j-1}$, $u$ and $v$. Further, Condition 1 holds with respect to $w$.

Since the algorithm yields tuples only when $w' = \varepsilon$ (Line 10), it follows that all produced tuples satisfy the lemma's conditions.

The initial elements inserted in Line 1 and Line 4 satisfy these properties trivially. Likewise, the element inserted in Line 7 satisfies the conditions: by the check in Line 6, some non-empty prefix of $w$ is a suffix of some $a \in R_t$. Choosing $u$ such that $u \circ w_{\leq l} = a$ and setting $v = \varepsilon$, one can check directly that all required conditions hold.

Assume inductively that all elements already in $S$ satisfy the conditions. We show that any new element added in the loop Lines 8–16 also satisfies the conditions. When Line 9 pops $(x, w', s, p)$ from $S$, we know by the induction hypothesis that there exist $u$ and $v$ such that $u \circ w_{1:|w|-|w'|} \circ v \in R(x)$. The check in Line 10 ensures $w' \neq \varepsilon$, hence $v = \varepsilon$ and not both $s = p = w$.

If Line 12 holds, then $w'$ is a prefix of some $a \in R_t$. In Line 13 we add $(x \circ t, \varepsilon, s, w')$. Let $a = w' \circ v$, then $u \circ w \circ v \in R(x \circ t)$. All required conditions now hold:

- By construction, $u \circ w \circ v \in R(x \circ t)$.

- $j = |w| + 1$ since $w' = \varepsilon$.

- Since $x$ satisfies Condition 1, so does $x \circ t$.

- Condition 2 does not apply.

- Condition 3 holds by the inductive hypothesis.

- Condition 4 holds since $t$ consumes $w' \circ v$.

Next, consider the element added in Line 16. By Line 14, $w'_{\leq l}$ is the largest prefix of $w'$ that matches $t$. Therefore, the element $(x \circ t, w'_{>l}, s, \text{None})$ added in Line 16 satisfies all conditions:

- By construction, $u \circ w_{1:|w|-|w'|} \circ w'_{\leq l} \in R(x \circ t)$.

- We may take $j$ such that $w'_{>l} = w_{j:|w|}$.

- Since $x$ satisfies Condition 1, so does $x \circ t$.

- Condition 2 does not apply.

- Condition 3 holds by the inductive hypothesis.

- Condition 4 holds since $p = \text{None}$ and $v = \varepsilon$.

Thus, all added elements satisfy the conditions, completing this direction of the proof.

**All tuples** Now let $(x, s, p)$ satisfy the lemma's conditions. If $s = p = w$, then $|x| \models 1$ and $x^{(1)} = x^{(-1)}$ consumes $u \circ w \circ v$. Thus, $w$ lies in the prefix language of the suffix language of $x$, and Line 3 inserts $(x, \varepsilon, w, w)$ into $S$. It is therefore yielded in Line 10.

Further, if $s \neq \varepsilon$, then by Condition 3, $x^{(1)}$ consumes $u \circ s$. Hence when $t = x^{(1)}$ in Line 2, Line 5 inserts $(x^{(1)}, w_{>|s|}, s, \text{None})$ into $S$.

Inductively, suppose $(x^{(1)}x^{(2)} \ldots x^{(i-1)}, w', s, \text{None})$ has been added to $S$, where $w'$ is the remaining suffix after consuming the prefix handled by $x^{(1)} \ldots x^{(i-1)}$ and $1 \leq i < |w|$. When $t = x^{(i)}$ in Line 11, the value of $l$ is precisely the length of the prefix of $w'$ consumed by $x^{(i)}$. Therefore Line 14 adds $(x^{(1)}x^{(2)} \ldots x^{(i)}, w'_{>l}, s, \text{None})$ as required.

Now, if $p = \text{None}$, a similar argument shows that $(x, \varepsilon, s, \text{None})$ is added to $S$, knowing that $(x^{(1)}x^{(2)} \ldots x^{(-2)}, w', s, \text{None})$ was added previously, since $x^{(-1)}$ fully consumes the remaining suffix $w'$ by the conditions.

Finally, we know that $(x^{(1)}x^{(2)} \ldots x^{(-2)}, p, s, \text{None})$ gets added to $S$, since the lexemes $x^{(1)}, \ldots, x^{(-2)}$ consume all of $w$ except $p$. Because $p$ lies in the prefix language of $x^{(-1)}$, condition Line 12 holds when $t = x^{(-1)}$. Thus, Line 12 inserts $(x, \varepsilon, s, p)$ into $S$.

This element is eventually popped and yielded in Line 10, completing the proof. $\square$

**Lemma 3.** *Given a Unicode string with holes $w_1 \square \ldots \square w_n$, Algorithm 4 returns* exactly *all lexeme sequences $x_1 \square x_2 \square \ldots \square x_m$ such that the following conditions hold: For any lexeme sequences $y_1, y_2, \ldots, y_{m-1}$, there exists a $w' \in R(w_1 \square \ldots \square w_n) \cap R(x_1 y_1 x_2 y_2 \ldots y_{m-1} x_m)$ such that $lex(w) = x_1 y_1 x_2 y_2 \ldots y_{m-1} x_m$, and each full string $w_i$ is consumed by a single lexeme sequence $x_j$ for some $1 \leq j \leq m$.*

*Proof.* We first show that any lexeme sequence returned by the algorithm satisfies the condition.

**No other sequences** First, let $x_1 \square x_2 \square \ldots \square x_m$ be any lexeme sequence returned by the algorithm. By construction, for each chunk $w_i$, there exists a tuple $(x'_i, s_i, p_i)$ returned by Algorithm 3 such that $x'_i$ is the lexeme sequence for chunk $w_i$. By Lemma 2, there exist strings $u_i$ and $v_i$ such that $u_i w_i v_i \in R(x'_i)$. Because of the yield (Line 31), we know that $x_i$ is either equal to $x'_j$ for some $j$, or is formed by merging $x'_j$ and $x'_k$ for some $j < k$, where the latter can only

happen if the intersection as given in Lines 13–20 is non-empty. Looking closely at this intersection, it can be easily determined that it implies the existence of strings $s_j, \ldots s_{k-1}$ such that $u_j w_j s_j w_{j+1} s_{j+1} \ldots s_{k-1} w_k v_k \in R(x_i)$. We set $w_i' = u_j w_j v_j$ for chunks that were not merged and to $w_i' = u_j w_j s_j w_{j+1} s_{j+1} \ldots s_{k-1} w_k v_k$ for merged chunks. Line 6 implies that $u_1 = v_n = \varepsilon$, implying that the concatenation $w_1' \square w_2' \square \ldots \square w_m' \in R(w_1 \square \ldots \square w_n)$. Further, we find that, since $w_i'$ is exactly consumed by $x_i$, $w_1' \square w_2' \square \ldots \square w_m' \cap R(x_1 y_1 x_2 y_2 \ldots y_{m-1} x_m) \neq \varnothing$ for any choice of lexeme sequences $y_1, y_2, \ldots, y_{m-1}$. Now, Corollary 1 and Assumption 2 shows that $\text{lex}(w) = x_1 y_1 x_2 y_2 \ldots y_{m-1} x_m$ for any $w \in R(w_1' \square \ldots \square w_m') \cap R(x_1 y_1 x_2 y_2 \ldots y_{m-1} x_m)$. Here, Assumption 2 is necessary to ensure that when we fill the $i$-th hole with an element from $R(y_i)$, we do not change the lexing of the surrounding lexemes sequences $x_i$ and $x_{i+1}$.

**All sequences** Now, let $x_1 \square x_2 \square \ldots \square x_m$ be any lexeme sequence that satisfies the condition. We show that the algorithm will eventually return $x_1 \square x_2 \square \ldots \square x_m$. Once again, we first define $w_i'$ as $w_j \square \ldots \square w_k$ if $x_i$ is the lexeme sequence that consumes chunks $w_j, w_{j+1}, \ldots, w_k$. Now, the characterization presented in Lemma 2 and the simple observation that Algorithm 4 uses a cross-product (Line 5) and returns lexeme sequences of all possible mergings (Lines 24–31) directly implies that $x_1 \square x_2 \square \ldots \square x_m$ will eventually be returned by the algorithm. $\square$

This characterization allows us to prove soundness and completeness of our lexing approach. Both of these proofs are one-liners that directly follow from Lemma 3. Importantly, completeness implies that if a proposed token by an LLM leads to an updated output that can be lexed to a valid lexeme sequence, we will accept the token. Further, soundness implies that if we accept a proposed token by the LLM, there exists a completion of the current output that can be lexed to a valid lexeme sequence.

**Theorem 4** (Soundness). *Given a partial Unicode string $\mathbf{w} = w_1 \square w_2 \square \ldots \square w_n$ with infilling regions, if Algorithm 4 returns a lexeme sequence that can be completed to a valid word in the language, then there exists a completion $\mathbf{w}'$ of $\mathbf{w}$ such that $\text{lex}(\mathbf{w}')$ is a valid word in the language.*

**Theorem 5** (Completeness). *Given a partial Unicode string $\mathbf{w}$ with infilling regions, if there exists a completion $\mathbf{w}'$ of $\mathbf{w}$ such that $\text{lex}(\mathbf{w}')$ is a valid word in the language, then Algorithm 4 will return a lexeme sequence that can be completed to a valid word in the language.*

## D EXPERIMENTAL DETAILS, ABLATIONS AND CASE STUDY

In this section, we provide additional details about the implementation, hyperparameters, datasets, runtime overhead, an ablation on the number of diffusion steps, and a case study.

### D.1 IMPLEMENTATION

**Overview** Our implementation is written in around 7000 lines of Python and 5500 lines of Rust. The main logic, concerning LLM sampling and CFG and DFA construction, is written in Python, with the more computationally expensive formal language operations, such as Algorithms 2 and 3, implemented in Rust, compiled as Python bindings. Several low-level formal language operation implementations are inspired by the educational Python implementations by Romero (2021).

**Grammars** Our C++ grammar covers a comprehensive but not complete subset of C++, with all features used in the canonical solutions of the test set implemented, but advanced features like template functions and user-defined classes are not supported. Moreover, we disallow the insertion of multi-line comments inside function bodies, as this allows the model to generate arbitrary and broken code that is syntactically valid as long as it is finally wrapped in the multi-line comment delimiters. We further restrict models in the MRI setting to not generate additional function signatures and bodies to prevent the generation of additional main functions or test cases.

We preprocess all model outputs by marking word boundaries with special $\langle$ and $\rangle$ tokens that do not appear in the original text and are never generated by the model[1]. For example, the string `int main()` is converted to $\langle$`int`$\rangle$ $\langle$`main`$\rangle$`()`. This enables us to check for such word boundaries inside

---

[1]In particular, we use the bytes `\x02` and `\x03`

the grammar, i.e., being able to distinguish whether white-space was present between symbols even after it is stripped in the lexing process.

The JSON schema grammars are obtained dynamically based on the JSON Schema for each task. We recursively build up the grammar based on the provided specification. For SMILES, we implement the specification described by Apodaca (2020), which is a more precise and efficient variant of prior specifications (Weininger, 1988; Blue Obelisk Project and OpenSMILES Community, 2025).

**Grammar Prompting**    For the grammar prompting ablation, we append the grammar in a human readable formatting into the prompt, i.e., without normalization and using descriptive symbol names. For JSON, the ad-hoc generated terminal names are replaced by a string representation of the respective regular expression, symbol names are $S1$ to $Sn$. An example for SMILES is shown in Figure 18

### D.2    MODELS AND HYPERPARAMETERS

All methods were run four times, with seeds $0$ to $4$, and we report the averaged results in all tables. We report the maximum among Van., Con.$^-$, and Con. decoding with **boldface**. We underline all results where the confidence interval of the improvement over the given method is not positive at $95\%$. We limit the amount of generated tokens to 256 and time out if the generation does not complete after 300 seconds. We run model inference on NVIDIA RTX A6000 GPUs.

**Sampling algorithms and temperature**    All MRI models were sampled with temperature 1 and greedy decoding. The diffusion models are sampled with a temperature of 0.2. To pick a token from the diffusion models distribution, we use the `entropy` algorithm for the DREAM 7B based models, DREAM 7B, DREAMCODER 7B, and DIFFUCODER 7B, and `low confidence` for the LLADA 8B model, as recommended by the model developers.

**Diffusion steps**    Diffusion language models can be run with a varying number of diffusion steps, determining how many tokens are sampled from a single model inference (Ye et al., 2025; Nie et al., 2025). Lower numbers of steps imply more tokens being sampled from each inferred distribution, which in turn is updated less frequently. One of the key benefits of diffusion language models is to exploit this ability, resulting in overall faster decoding. At the same time, higher numbers of steps are usually associated with increased accuracy on the requested task, as the model can adapt its distribution more frequently to newly inserted tokens. When not explicitly stated otherwise, the diffusion models are run with 32 diffusion steps. Our choice of step size 32 represents a trade-off between speed and accuracy.

In each diffusion step, model inference is run once on the current state of the partially filled context window. Afterwards, $\frac{n}{k}$ tokens are sampled from the distribution according to the respective algorithms (`low confidence` or `entropy`) and replace mask tokens in the context window. While unconstrained decoding allows sampling all $\frac{n}{k}$ tokens in parallel, during constrained decoding, we iteratively sample single token-index pairs from this distribution, with rejections leading to masking out the rejected token-index pair and resampling. When a token is accepted, we remove the token's index from the distribution. After $\frac{n}{k}$ tokens have been accepted, we run model inference again.

### D.3    DATASETS

**C++**    We leverage the C++ translation of HumanEval in the HumanEval-X dataset (Zheng et al., 2023). It contains 164 instances of simple programming problems and canonical solutions written by humans. For the MRI tasks, we remove between 1 and 3 randomly sized spans of 5 to 100 characters from these canonical solutions, generating one MRI task per instance in the original dataset. If we end up with insufficient remaining characters after removing the required number of spans, we resample sizes and positions up to 3 times, aborting if we do not find a valid removal. Additionally, we remove 5 human-written solutions that are not valid according to our implementation of the syntax, i.e., because they contain multi-line comments or additional helper functions. This results in three MRI datasets of 159, 156, and 143 samples in 1-MRI, 2-MRI and 3-MRI respectively. An example prompt for an instance from the dataset is presented in Figure 9. For DLM, we use all 164

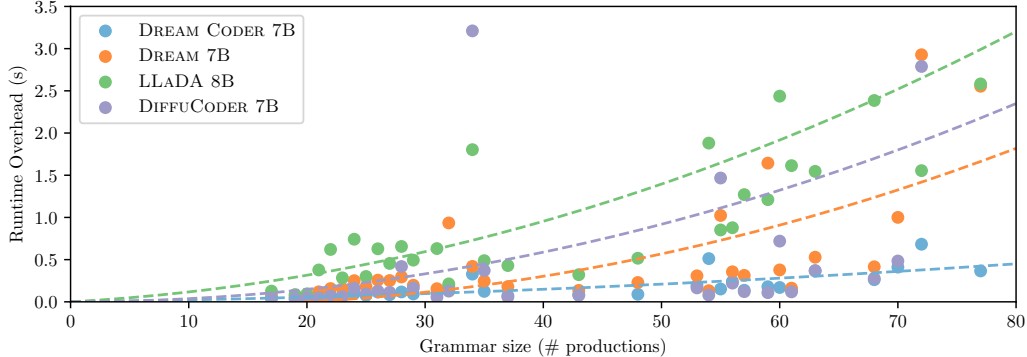

Figure 7: The median runtime overhead of Con. grouped by size of the grammars in JSON, with a 2-degree polynomial fitted against the individual points. A clear increase can be seen, although it ranges only between 0 and 3 seconds.

tasks of the original dataset and extract the comment before the function as an instruction for the model. An example prompt is shown in Figure 10.

We check the functional correctness in both settings by checking whether all test cases in the dataset pass with the model-generated solution.

**JSON Schema**    We extend the JSON-Schema dataset by NousResearch (2024). Concretely, the dataset originally contains a unique schema per task. We clean the schemas by disallowing properties other than specified on the top level and repairing instances that accidentally do not require any fields. We then extend the dataset by sampling GEMINI-2.5-PRO for 10 inputs and completions for each schema. We filter these samples in three ways to ensure high quality.

First, we filter the resulting extracted outputs for syntactic validity according to the schema and discard invalid generations. Second, we require GEMINI-2.5-PRO to be able to solve the task, i.e., the model must generate a valid JSON object that passes the schema validation if it is only given the input and the schema. Third, we perform fuzzy matching to deduplicate the resulting samples. This process results in 272 instances. The prompts used for generation and verification are shown in Figure 13 and Figure 14. An example prompt for this task is shown in Figure 11.

We evaluate functional correctness on this dataset by checking for exact equality between a normalized JSON dump of the golden solution and the model-generated solution.

**SMILES**    To create a benchmark for SMILES, we query GEMINI-2.5-PRO to generate pairs of descriptions of molecules and their SMILES notation. Again, we perform three filtering steps to ensure high quality. First, we verify that the generated molecule is valid using the Rdkit library (Landrum et al., 2025). Second, we ensure the model can generate the correct SMILES string for the molecule if it is only given the description. Third, we filter out duplicates using fuzzy matching. This results in 167 pairs of descriptions and SMILES strings. Prompts for this generation procedure are shown in Figure 15 and Figure 16. An example prompt for this task is shown in Figure 12.

To check the functional correctness of the model-generated molecule, we parse it using Rdkit and check the equivalence to the molecule generated by GEMINI-2.5-PRO in canonical representation.

## D.4    ADDITIONAL EXPERIMENTS

**Runtime overhead**    For all experiments in §4, we measure the runtime of our constraining method and unconstrained decoding. We present a detailed comparison in Tables 3 and 4. We further measure the average number of rejections per sample.

In MRI we compare time per token, as constrained decoding often rejects finalizing the current output, thus making completions longer and finalization times incomparable. The median runtime

Table 3: Median overhead per token for different infilling settings in milliseconds and percent increase over unconstrained generation. Larger models with higher inference time experience a lower slowdown due to constraining. More infilling regions also increase constraining overhead.

| #Regions | 1-MRI | 2-MRI | 3-MRI |
|---|---|---|---|
| CODEGEMMA 7B | $3.1_{\uparrow 47\%}$ | $4.1_{\uparrow 63\%}$ | $6.4_{\uparrow 99\%}$ |
| STARCODER2 7B | $3.3_{\uparrow 59\%}$ | $5.5_{\uparrow 98\%}$ | $9.7_{\uparrow 190\%}$ |
| DEEPSEEK C. 1.3B | $3.6_{\uparrow 158\%}$ | $5.8_{\uparrow 245\%}$ | $11.8_{\uparrow 557\%}$ |
| DEEPSEEK C. 6.7B | $3.0_{\uparrow 58\%}$ | $4.6_{\uparrow 90\%}$ | $7.7_{\uparrow 153\%}$ |
| DEEPSEEK C. 33B | $3.1_{\uparrow 13\%}$ | $4.3_{\uparrow 19\%}$ | $6.5_{\uparrow 28\%}$ |

Table 4: Median time difference per completion for different diffusion models in seconds, and the overhead over the original completion in percent. When the completion aborts pre-emptively, as no valid completion is sampled from the model, speed-ups are possible.

| Model | C++ | JSON | SMILES |
|---|---|---|---|
| DREAM 7B | $1.1_{\uparrow 36\%}$ | $0.4_{\uparrow 20\%}$ | $0.0_{\uparrow 0\%}$ |
| DREAMC. 7B | $7.8_{\uparrow 190\%}$ | $0.1_{\uparrow 5\%}$ | $0.0_{\uparrow 1\%}$ |
| LLADA 8B | $-1.0_{\downarrow 19\%}$ | $0.5_{\uparrow 9\%}$ | $0.0_{\uparrow 1\%}$ |
| DIFFUC. 7B | $2.2_{\uparrow 74\%}$ | $0.1_{\uparrow 6\%}$ | $0.0_{\uparrow 2\%}$ |

overhead of constrained decoding is $125\%$, where the overhead on the small DEEPSEEK CODER 1.3B is higher ($320\%$) than on the 7B model ($100\%$) and DEEPSEEK CODER 33B ($20\%$). This is both due to the lower inference time of smaller models, and due to smaller models making more mistakes, with the average number of rejections increasing from $8.8$ per instance on 33B, over $9.7$ for 7B to $10.5$ in 1.3B. Moreover, more infilling regions are more difficult, leading to more rejections, growing from $4.7$ on 1-MRI to $14.1$ in 3-MRI. This increases the overhead from $67\%$ to $205\%$ respectively.

For DLM, we compare the total runtime to finish the diffusion decoding process. The average completion overhead is only $30\%$, but varies strongly between domains. We observe both speed-ups of up to $19\%$, for LLADA 8B on C++, where many decodings are preemptively aborted, and slowdowns of up to $190\%$, for DREAMCODER 7B on the same dataset.

We further analyze the runtime overhead based on grammar size and infilling regions. We first analyze the runtime overhead per size of the normalized grammar, as the normalized grammar is the basis for our intersection algorithm. The normalized grammar sizes of C++ and SMILES are 167 productions and 46 productions respectively. As shown in Table 4, this corresponds to a median runtime overhead of $2.5\,\mathrm{s}$ and $0.0\,\mathrm{s}$ respectively, indicating a strong correlation to the grammar size. For a closer analysis, we plot the median runtime overhead for each of the JSON tasks in Figure 7, where each task has an individual grammar, and regress a 2-degree polynomial against the data. An increase in runtime, according to the asymptotic complexity described in §3.2 is visible, but the constant appears very small. For an analysis of the impact of infilling regions, we refer to our results of runtime overhead for increasing number of infilling regions in MRI Table 3, where more infilling regions increase runtime from $3.2\,\mathrm{s}$ to $8.4\,\mathrm{s}$.

**Ablation on diffusion steps**  We evaluate our method on common diffusion step numbers, from 16 to 256, where the lowest setting 16 implies that a single inference step inserts $\frac{256}{16} = 8$ tokens at once, while the highest setting 256 implies that every inference step inserts only a single token.

We present the results of this ablation on DREAM 7B in Table 5 and demonstrate that our method consistently improves syntactic correctness in all settings by on average $14\%$. Functional correctness on JSON also significantly increases by $1.2\%$, while the increase in C++ is $0.7\%$ and $0.5\%$ in SMILES. Moreover, the runtime overhead, shown in Table 6, decreases with the number of diffusion steps, from $14\% - 108\%$ down to $9\%$ or even a speed up of $3\%$.

Table 5: Percent syntactically and functionally correct generations for DREAM 7B based on varying number of diffusion steps. Our method consistently increases syntactic correctness in all settings, even when model accuracy increases with step sizes.

|  | #Steps | C++ | | | JSON | | | SMILES | | |
|---|---|---|---|---|---|---|---|---|---|---|
|  |  | Van. | Con.⁻ | Con. | Van. | Con.⁻ | Con. | Van. | Con.⁻ | Con. |
| Syntax | 16 | 8.1 | 20.3 | **99.2** | 7.3 | 24.4 | **100.0** | 41.1 | 80.5 | **99.7** |
|  | 32 | 40.5 | 58.7 | **99.4** | 22.4 | 44.9 | **100.0** | 67.5 | 93.7 | **99.4** |
|  | 64 | 60.1 | 74.7 | **99.8** | 67.4 | 73.2 | **100.0** | 79.2 | 94.9 | **100.0** |
|  | 128 | 81.1 | 90.7 | **100.0** | 90.2 | 94.0 | **100.0** | 80.1 | 95.8 | **100.0** |
|  | 256 | 98.2 | 98.2 | **100.0** | 95.2 | 98.2 | **100.0** | 80.7 | 93.4 | **100.0** |
| Functional | 16 | 1.4 | 2.7 | **4.9** | 1.5 | 2.3 | **3.4** | 0.6 | **1.1** | 1.1 |
|  | 32 | 6.6 | 8.8 | **9.5** | 7.4 | 11.4 | **14.3** | 0.6 | **1.1** | 1.1 |
|  | 64 | 21.0 | 21.8 | **22.4** | 41.8 | 42.1 | **42.8** | 2.4 | **3.0** | 3.0 |
|  | 128 | **24.5** | 23.9 | 24.1 | 50.7 | **51.5** | 51.5 | 3.1 | **4.0** | 4.0 |
|  | 256 | 34.1 | 34.1 | **34.8** | 54.8 | 54.8 | 54.8 | 4.9 | **5.2** | 5.2 |

Table 6: Time difference per completion for different step sizes on DREAM 7B diffusion, in seconds, and the percentual overhead over the original completion. For larger numbers of diffusion steps, overhead reduces from $14\% - 108\%$ down to $9\%$ or even a speedup of $1\%$.

| #Steps | C++ | JSON | SMILES |
|---|---|---|---|
| 16 | $1.7_{\uparrow 107\%}$ | $2.1_{\uparrow 108\%}$ | $0.0_{\uparrow 14\%}$ |
| 32 | $1.1_{\uparrow 36\%}$ | $0.4_{\uparrow 20\%}$ | $0.0_{-}$ |
| 64 | $0.6_{\uparrow 18\%}$ | $0.1_{\uparrow 4\%}$ | $-0.2_{\downarrow 3\%}$ |
| 128 | $0.4_{\uparrow 10\%}$ | $0.2_{\uparrow 4\%}$ | $-0.4_{\downarrow 3\%}$ |
| 256 | $0.8_{\uparrow 9\%}$ | $-0.1_{\downarrow 1\%}$ | $0.2_{\uparrow 1\%}$ |

Table 7: When infilling between 1 and 3 missing lines, our method consistently improves syntactic and functional correctness. Shown below the results of MRI-L under standard decoding (Van.), constrained decoding (Con.⁻), and completing partially completed outputs (Con.).

|  | Model | 1-MRI-L | | | 2-MRI-L | | | 3-MRI-L | | |
|---|---|---|---|---|---|---|---|---|---|---|
|  |  | Van. | Con.⁻ | Con. | Van. | Con.⁻ | Con. | Van. | Con.⁻ | Con. |
| Syntax | STARCODER2 7B | 84.0 | 95.4 | **98.3** | 78.5 | 91.3 | **96.5** | 63.5 | 86.8 | **95.7** |
|  | CODEGEMMA 7B | 94.8 | 95.7 | **98.8** | 83.8 | 93.9 | **97.3** | 70.1 | 91.3 | **96.9** |
|  | DEEPSEEK C. 1.3B | 62.3 | 69.1 | **87.2** | 30.8 | 44.5 | **75.8** | 23.8 | 39.4 | **76.7** |
|  | DEEPSEEK C. 6.7B | 64.0 | 74.5 | **95.7** | 29.6 | 52.6 | **93.3** | 24.1 | 49.8 | **89.6** |
|  | DEEPSEEK C. 33B | 65.2 | 71.2 | **92.8** | 30.0 | 45.0 | **87.3** | 22.9 | 39.1 | **84.4** |
| Functional | STARCODER2 7B | 69.5 | 71.8 | **72.0** | 48.6 | **49.8** | **49.8** | 28.5 | **31.9** | **31.9** |
|  | CODEGEMMA 7B | **82.2** | 79.4 | 79.4 | 53.4 | 55.8 | **55.8** | 34.5 | 40.5 | **40.6** |
|  | DEEPSEEK C. 1.3B | 0.9 | 5.2 | **8.7** | 1.1 | 2.0 | **2.6** | 0.5 | 1.5 | **1.8** |
|  | DEEPSEEK C. 6.7B | 0.6 | 8.8 | **13.7** | 1.2 | 2.7 | **3.7** | 0.0 | 0.8 | **1.1** |
|  | DEEPSEEK C. 33B | 0.6 | 5.9 | **10.1** | 1.4 | 3.4 | **4.0** | 0.0 | 1.4 | **2.1** |

**Removing entire lines in MRI** The MRI setting is based on the tasks proposed by Bavarian et al. (2022). In this work, two settings are suggested: removing random spans and removing entire lines of code. In §4, we presented the results for MRI when removing random spans. Here, we explore the alternative of removing lines, which we call MRI-L. Crucially, lines are often semantically coherent and self-contained, thus posing a different challenge. Equivalenty to MRI, we introduce 1-MRI-L, 2-MRI-L and 3-MRI-L, where we remove between one and three lines from the code. If lines are adjacent, we merge them into a single span to complete.

We present the results in Table 7. We observe that this task appears more difficult for the LLMs, contrasting the observations by Bavarian et al. (2022). Concretely, model performance for the DEEPSEEK CODER family drops to near 1%. Meanwhile, matching the results observed in §4 for random spans, in all but one setting our constrained decoding improves model syntactic and semantic correctness, often significantly.

## D.5 COMPARISON TO DINGO

In this section we compare the performance of our method to the method DINGO proposed by Suresh et al. (2025). This method is specifically designed for diffusion language models, but can only enforce regular constraints. As such, it is less general than our method, which can enforce context-free constraints. Since the method is not openly accessible, we attempt matching their generation settings and datasets here and compare to the results by them.

**Datasets** Suresh et al. (2025) propose two datasets for constrained decoding.

The first dataset JSON-NOUS is similar to our dataset JSON: The model is provided with a natural language text and asked to extract data matching a provided JSON schema. The underlying dataset contains the 100 tasks of NousResearch (2024) on which our dataset JSON is based. As opposed to our implementation, they count as syntactically correct any output that is generally valid JSON, and functionally correct any output that adheres to the JSON-Schema.

The second dataset, GSM8K-SYMBOLIC (Mirzadeh et al., 2025), provides the LLM with a symbolic representation of arithmetic tasks, based on the GSM8K dataset proposed by Cobbe et al. (2021)[2]. The LLM should generate an arithmetic expression that uses basic arithmetic operators and specific variables, and that matches the ground-truth solution. An example is provided in Figure 17. We exclude the tasks whose golden solutions require conditionals (x if y else z) and interpret int(x) and frac(y,z) as x and y/z respectively. Syntactically correct outputs must have valid arithmetic expressions in the last occurring pair of << and >>. Functionally correct outputs provide an equivalent formula as the golden solution, based on an assessment by Z3 (De Moura and Bjørner, 2008).

---

[2]It should be pointed out however, to avoid confusion, that this evaluation utilizes the dataset produced by Mirzadeh et al. (2025) in a very different way than suggested by them.

Table 8: On JSON-NOUS, our method (Con.) achieves the same performance as DINGO. On GSM8K-SYMBOLIC, DINGO slightly outperformsn Con..

| | Model | JSON-NOUS | | | | GSM8K-SYMBOLIC | | | |
|---|---|---|---|---|---|---|---|---|---|
| | | Van. | DINGO | Con.$^-$ | Con. | Van. | DINGO | Con.$^-$ | Con. |
| Syn. | LLADA 8B | 93.0 | **100** | 88.5 | **100.0** | 72.0 | 100 | 87.0 | **100.0** |
| | DREAM 7B | 89.0 | **100** | 91.0 | **100.0** | 65.0 | 100 | 69.0 | **100.0** |
| Func. | LLADA 8B | 86.2 | **100** | 87.5 | **100.0** | 24.7 | **29** | 27.7 | 27.7 |
| | DREAM 7B | 85.0 | **100** | 90.0 | **100.0** | 25.0 | **34** | 25.0 | 25.0 |

Table 9: Pre-procesing time (Pre$_X$) and time difference per completion of DINGO and our method (Con.). We observe that our method has a similar runtime overhead as DINGO while requiring no preprocessing. Notably, preprocessing is done once per schema, which implies once per task on the JSON-NOUS dataset.

| | JSON-NOUS | | | | GSM8K-SYMBOLIC | | | |
|---|---|---|---|---|---|---|---|---|
| Model | Pre$_{\text{DINGO}}$ | Pre$_{\text{Con.}}$ | DINGO | Con. | Pre$_{\text{DINGO}}$ | Pre$_{\text{Con.}}$ | DINGO | Con. |
| LLADA 8B | 13.2 | 0 | 0.1 | 0.3 | 32.1 | 0 | 0.1 | 0.3 |
| DREAM 7B | 11.9 | 0 | 0.1 | 0.1 | 37.0 | 0 | 0.1 | 0.2 |

**Grammars** Although both tasks require context-free grammars to be accurately solved, Suresh et al. (2025) disallow context-free parts of the JSON-schema language and hard-code a maximum nesting for arithmetic expressions to allow solving the tasks in these datasets with terms matched by regular expressions, provided in the Appendix of Suresh et al. (2025). Our implementation for JSON-NOUS constructs the same grammars as in our dataset JSON. For GSM8K-SYMBOLIC, we introduce a new grammar that allows the language model to first reason freely (allowing any characters), and constrains only the content of the last pair of << and >> to a simple, context-free language of arithmetic expressions with addition, subtraction, multiplication, exponentiation and modulo.

**Models and Parameters** We compare the performance of LLADA 8B and DREAM 7B with 64 diffusion steps, generation length of 128, 1 generation block and temperature 0.2 and sample 4 times to compute confidence intervals.

**Results** The syntactic and functional results are presented in Table 9. We observe that both our method Con. and DINGO correctly ensure perfect adherence to JSON-Schemas in the task JSON-NOUS. For GSM8K-SYMBOLIC, we observe that DINGO and Con. both enforce 100% syntactic correctness, but DINGO slightly outperforms Con. in functional correctness. This can be due to multiple reasons: The alignment of DINGO is closer to a deeply grammar-aligned solution as proposed by Park et al. (2024) than the grammar-constrained approach that our method uses. Further, DINGO hardcodes the maximum nesting of arithmetic expressions, thereby biasing the model towards correct solutions.

Comparing the runtime overhead of both methods, we observe that our method and DINGO achieve similar runtime overhead per sample, with both impacting generation by less than 0.3s. Notably, DINGO requires additional preprocessing for each grammar, which takes up to half a minute. In the case of JSON-NOUS, this implies additional preprocessing for every sample, as each JSON Schema defines its own grammar. Meanwhile, our method requires no preprocessing.

### D.6 CASE STUDY

For a qualitative evaluation, we manually inspect instances where unconstrained decoding fails and our constraining approach successfully corrects errors. We showcase three such examples in Table 10.

Table 10: Three examples demonstrating the impact of constrained decoding on DLM and MRI completion. Left are unconstrained completions (Van.) with problematic tokens highlighted in red , and right constrained completions (Con.⁻) with corrections highlighted in green , adapted for clarity. In (a), our method forces DREAM 7B to generate values of the correct type in a summary of a financial review for task #30 in our JSON dataset. In (b), generated by DREAMCODER 7B, our method prevents closing more parentheses than are opened when generating a SMILES molecule for task #166. In (c), our method forces DEEPSEEK C. 6.7B to add parentheses around a condition in an if-statement when writing a string processing function in task #150 of our C++ dataset.

| | Vanilla | Constrained |
|---|---|---|
| (a) | ```// summarize my financial review

{
  "capitalGains": "5210.5000" ,
  "interestIncome": "1340.25" ,
  "totalReturn": "4.5"
}``` | ```// summarize my financial review

{
  "capitalGains": 5210.5000 ,
  "interestIncome": 1340.25 ,
  "totalReturn": 4.5
}``` |
| (b) | ```// generate an allene with axial chirality

C1=CC1=CC(C(00)0)C(00)0 )C(00)0``` | ```// generate an allene with axial chirality

C1=CC1=CC(C(00)0)C(00)0``` |
| (c) | ```// separate the groups of nested parentheses

if (chr=='(')
{
    level+=1;
    current_paren+=chr;
}
else if chr==')'
{
    ...``` | ```// separate the groups of nested parentheses

if (chr=='(')
{
    level+=1;
    current_paren+=chr;
}
else if (chr==')')
{
    ...``` |

```
1  vector<string> numerical_letter_grade(vector<float> grades){
2      vector<string> out={};
3      for (int i=0;i<grades.size();i++)
4      {
5          if (grades[i]>=3.9999) out.push_back("A+");
6          if (grades[i]>3.7001 and grades[i]<3.9999) out.push_back("A");
7          if (grades[i]>3.3001 and grades[i]<=3.7001) out.push_back("A-");
8          if (grades[i]>3.0001 and grades[i]<=3.3001) out.push_back("B+");
9          if (grades[i]>2.7001 and grades[i]<=3.0001) out.push_back("B");
10         if (grades[i]>2.3001 and grades[i]<=2.7001) out.push_back("B-");
11         if (grades[i]>2.0001 and grades[i]<=2.3001) out.push_back("C+");
12         if (grades[i]>1.7001 and grades[i]<=2.0001) out.push_back("C");
13         if (grades[i]>1.3001 and grades[i]<=1.7001) out.push_back("C-");
14         if (grades[i]>1.0001 and grades[i]<=1.3001) out.push_back("D+");
15         if (grades[i]>0.7001 and grades[i]<=1.0001) out.push_back("D");
16         if  □ i]<=3.0001) out.push_back("B");
17         if (grades[i]>2.3001 and grades[i]<=2.7001) out.push_back("B-");
18         if (grades[i]>2.0001 and grades[i]<=2.3001) out.push_back("C+");
19         ...
```

(a) STARCODER2 7B exceeds the token limit in task #81 in 1-MRI.

| Vanilla | Constrained$^-$ | Constrained |
|---------|-----------------|-------------|
| C6CCCC1) | C6CCCC$\perp$)) | C6CCCC(c(c)(c)) |

(b) LLADA 8B leaves a single $\perp$ for completion in task #153 in SMILES.

Figure 8: Syntax errors may remain when the model has fewer tokens left to complete than would be required to fulfil the syntactic constraints. This can happen both in MRI (a), when the model exceeds the maximum number of generated tokens and in DLM (b), when the model has few mask tokens $\perp$ remaining.

**Preventing use of invalid types**   In Table 10a, DREAM 7B generates a summary of a financial review for task #30 in our JSON dataset. The schema requires three values of type float. However, the model attempts to generate these values as strings. By applying our constraining method, it can be determined that the strings are misplaced, not constituting one of the required values, and can not match the intended value type. All attempts at placing such strings are thus rejected during generation. Instead, our method forces the model to generate the values without inserting quotes, resulting in a valid and correct result.

**Preventing incorrect nesting**   In Table 10b DREAMCODER 7B generates an invalid SMILES molecule for task #166 by closing more parentheses than it opened. Since the CFG for SMILES correctly handles counting of nesting levels, attempts to generate the closing parentheses are rejected by our method. Instead, the model decides to end the generation.

**Preventing inadequate syntax**   In Table 10c, DEEPSEEK C. 6.7B uses conditions without parentheses in an if-statement when writing a string processing function in task #150 of our C++ dataset. This confusion may stem from the dominance of Python code in training data, which does not require parentheses in if-statements. However, this is invalid according to C++ syntax. Our method can correct this mistake successfully, resulting in a correct infilling.

# E   DISCUSSION

**Remaining syntax errors**   While our method achieves substantial improvements in syntactic correctness, using only Con.$^-$ still leaves a considerable gap until guaranteeing correctness. We attribute most of this gap to the overapproximation of allowing an arbitrary number of tokens to

fill regions in the partial output, as done in prior work (Beurer-Kellner et al., 2024; Ugare et al., 2024). In practice, the LLM is typically limited, i.e., in FIM and MRI it can only generate up to the user-defined maximum number of tokens, and in DLM it can only generate one token per mask $\perp$. Examples of this issue occurring are presented in Figure 8a for MRI, where the model exceeds the token limit of 256 tokens as it generates large amounts of unnecessary code, and in Figure 8b, where the DLM model needs to open several molecule branches in a single remaining token.

One approach to resolve this issue would be to accurately model the remaining number of tokens in our regular language construction. However, we observe in experiments that this significantly increases the size of the regular language, as it consequently needs to keep track of the number of inserted tokens. This drastically increases the size of the intersection language, rendering our current implementation impractical.

Another approach would be to train the model to signal requiring additional tokens. In MRI, this naturally occurs when the model does not generate an end-of-string token. For DLM, a special token could be added that is replaced with two mask tokens after sampling, increasing the size of the affected infilling region. Concurrent work by Wu et al. (2025) reports that such capabilities appear to generally improve model performance for code infilling.

Our chosen approach to mitigate the issue is to automatically fill in the output based on the formal language constraints (Con.). However, this solution cannot rely on the model's probability distribution to steer generation. Determining the most effective way to handle this limitation is an important topic for future work.

**Leveraging incremental parsing** While we take several steps to improve the efficiency of our method, it can still require a significant amount of time to determine the emptiness of the intersection language after each generated token. Future work may leverage the fact that the CFG for the intersection is fixed and the DFA is only updated using small modifications. This may lead to an approach for incrementally computing emptiness checks by reusing the results of the previous intersection computation. Other approaches to leverage the incremental nature of the parsing, similar to the approaches of Melcer et al. (2024); Ugare et al. (2024), and Mündler et al. (2025) would likely also be able to decrease the worst case and practical overhead of the constraining method.

**Context-sensitive language features** While our method is designed for context-free languages, an interesting future direction would be extensions to handle more powerful language classes, such as context-sensitive languages. Similar to Melcer et al. (2024) and Ugare et al. (2024), simple context-sensitive syntactic features can likely be handled by preprocessing through adequate lexers. Beyond syntactic features, prior work suggested leveraging more semantic insights, such as type systems (Mündler et al., 2025), for constructing more powerful constraint systems. Type checkers with typed holes (Omar et al., 2019) could be leveraged to achieve such systems.

# F PROMPTS

In this section, we detail all prompts used for the respective models and tasks.

**MRI** Since models used for FIM and MRI tasks are not instruction-fine-tuned, we provide the model with the raw code, including only a comment above the function to guide the model for completions. We use the standard templating suggested for each model to format the prompt for FIM and MRI completion. If MRI is not supported explicitly, we emulate it by inserting <TODO> into the remaining infilling regions and repeatedly prompting the model for FIM completion on the first infilling region. In order to prevent the models from generating main methods and tests in the MRI setting, we add a main method at the end of the context that is marked with a TODO comment. An example prompt for the 2-MRI setting is provided in Figure 9.

**DLM** The models used for DLM tasks are instruction-fine-tuned, allowing us to specify the completion intent in natural language. We provide a general description of the task in the system prompt and the specific task content as the first user prompt. The assistant response is prefilled with the start of a code fence and, in the case of C++, with the necessary header declarations and function

signature to ensure results can be extracted and tests can be executed correctly. The prompts for C++, JSON and SMILES tasks are presented in Figure 10, 11 and 12 respectively.

**Benchmark generation prompts**  As outlined in Appendix D, we generate the JSON and SMILES dataset synthetically by prompting GEMINI-2.5-PRO. We provide the used prompts for generation and validation of the generated samples in Figures 13–16.

**Miscellaneous Prompts**  The prompt for the GSM8K-SYMBOLIC experiment for DINGO is shown in Figure 17. An example prompt for the grammar prompting (G.P.) is shown in Figure 18.

```
1  region 0
2      /*
3      From a given vector of integers, generate a vector of rolling maximum element found
4      until given moment in the sequence.
5      >>> rolling_max({1, 2, 3, 2, 3, 4, 2})
6      {1, 2, 3, 3, 3, 4, 4}
7      */
8      #include<stdio.h>
9      #include<vector>
10     using namespace std;
11     vector<int> rolling_max(vector<int> numbers){
12         vector<int> out;
13
14 region 1
15         for (int i=0;i<numbers.size
16
17 region 2
18         return out;
19 }
20
21 int main(){
22     // TODO
23 }
```

Figure 9: Example prompt for the 2-MRI task #1. The intial comment and function signature in blue are derived from the dataset prompt, and the remaining code snippets in green are the remainders of the canonical solution with two randomly removed spans. We append a stub main function to prevent the model from attempting to generate a main function of its own.

```
1  system
2      You are an expert in C++ programming. Solve the given problem by writing solution
3       code in C++.
4      When answering, insert the solution code in a ```cpp...``` block. Do neither include
5       test cases not a main function.
6
7  user
8      Check if in given vector of numbers, are any two numbers closer to each other than
9      given threshold.
10     >>> has_close_elements({1.0, 2.0, 3.0}, 0.5)
11     false
12     >>> has_close_elements({1.0, 2.8, 3.0, 4.0, 5.0, 2.0}, 0.3)
13     true
14
15 assistant
16     ```cpp
17     #include<stdio.h>
18     #include<vector>
19     #include<math.h>
20     using namespace std;
21     #include<algorithm>
22     #include<stdlib.h>
23     bool has_close_elements(vector<float> numbers, float threshold){
```

Figure 10: Example prompt for the C++ task #1. The system prompt in black is fixed, whereas the user prompt in blue is extracted from the comment preceding the function and the assistant response is prefilled with a codefence, and in green, headers, and the function signature of each task.

```
1
2  system
3      You are a helpful assistant that answers in JSON. Here is the JSON schema you must
4      adhere to:
5      <schema>
6      {
7          "type": "object",
8          "properties": {
9              "name": {
10                 "type": "string"
11             },
12             "email": {
13                 "type": "string"
14             },
15             "shippingAddress": {
16                 "type": "string"
17             }
18         },
19         "required": [
20             "name",
21             "email",
22             "shippingAddress"
23         ],
24         "additionalProperties": false
25     }
26     </schema>
27
28 user
29     We are registering 'Global Exports Ltd.' for your services. The main contact person
30     is Samantha Davis, and her corporate email is s.davis@globalexports.co.uk. All ship-
31     ments and correspondence should be directed to our headquarters: Global Exports Ltd.,
32     12 Business Park Road, Manchester, M1 1AB, United Kingdom. We are looking forward to
33     a fruitful partner ship and are particularly interested in your international ship-
34     ping rates.
35
36 assistant
37     ```json
```

Figure 11: Example prompt for the JSON task. The JSON schema in green is task-specific as well as the the user prompt in blue from which information should be extracted into the given schema. The system prompt and prefilled assistant response are fixed.

```
1
2  system
3      You are a specialized AI assistant that generates SMILES (Simplified Molecular Input
4      Line Entry System) strings from chemical descriptions. You will be given a textual
5      description of a chemical compound or a related task. Your goal is to produce the
6      most accurate and valid SMILES string representing that description.
7
8      Your Task:
9
10     Based on the provided "input" description, generate the corresponding SMILES string.
11
12     Output Requirements:
13
14     - Provide only the SMILES string as your output.
15     - Ensure the SMILES string is syntactically valid.
16     - Represent all specified chemical features accurately (atoms, bonds, rings,
17         aromaticity, charge, isotopes, stereochemistry).
18
19     Output:
20
21     - Provide only the smiles molecule as a raw string between triple backticks (```).
22     For instance:
23     ```smiles
24     C1=CC=CC=C1
25     ```
26
27 user
28     Propan-1-amine, a primary amine with a three-carbon straight chain and the amino
29     group on the first carbon.
30
31 assistant
32     ```smiles
```

Figure 12: Example prompt for the SMILES task. The user prompt in blue varies per task.

```
1   user
2       Your goal is to create challenging and diverse `JSON Schema` problems. You are
3       given a JSON schema that describes a specific schema for a JSON problem.
4
5       You should generate **{num_samples}** JSON benchmark samples based on the
6       provided schema. A benchmark sample consists of a natural language description
7       describing how the JSON schema should be filled out, along with a JSON object
8       that adheres to the schema.
9
10      For each sample, provide a JSON object with the following structure:
11
12      ```json
13      {{
14          "input": "A natural language description of how the JSON schema should be
15              filled out. The input should be a natural query that a user might ask an
16              LLM. The input will be given to the LLM as a prompt, along with the JSON
17              schema. Based on this input, the LLM should generate a JSON object that
18              adheres to the schema.",
19          "output": "A JSON object that adheres to the provided schema. The output
20              should be a valid JSON object that matches the schema and reflects the
21              input description."
22      }}
23      ```
24
25      **Guidelines for generating samples:**
26
27      - **Variety**: Describe a wide range of scenarios that can be expressed using
28          the JSON schema. Ensure that the samples cover a wide range of possible
29          scenarios, and make them sound natural and plausible.
30      - **Difficulty**: User queries can and should contain distracting information
31          and longer backgrounds.
32      - **Realism**: Test cases should reflect plausible scenarios where the JSON
33          schema would be used.
34      - **Reference**: Do not reference the JSON schema in the input description. The
35          input should be a natural query that a user might ask an LLM. It should not
36          reference JSON at all.
37
38      JSON Schema:
39      {schema}
40
41      Example Input (Do not use this in your samples):
42      {input_query}
43
44      Example Output (Do not use this in your samples):
45      {output_query}
```

Figure 13: Prompt used to generate additional JSON Schema samples for the JSON task using GEMINI-2.5-PRO. Several samples were generated at the same time to increase diversity.

```
1   user
2       You are a JSON Schema assistant. You will be given a textual description of how
3       a JSON schema should be filled out. Your task is to generate a JSON object that
4       adheres to the provided schema.
5
6       Your Task:
7       - Analyze the textual task.
8       - Construct a JSON object that correctly implements the task based on the
9           provided schema.
10
11      The JSON object should be a valid JSON object that matches the schema and
12      reflects the input description.
13
14      Output:
15      - Provide only the JSON object as a raw string between triple backticks
16          (```json). Ensure the JSON object satisfies the JSON schema. For instance:
17      ```json
18      {{
19      "key": "value",
20      "number": 42,
21      "array": [1, 2, 3]
22      }}
23      ```
24
25      Json Schema:
26      {schema}
27
28      Description:
29      {input_query}
```

Figure 14: Prompt used to verify additional JSON Schema samples for the JSON task using GEMINI-2.5-PRO.

```
1   user
2       You are a specialized AI assistant tasked with generating benchmark samples for
3       SMILES (Simplified Molecular Input Line Entry System) string generation. Your
4       goal is to create diverse and accurate chemical structure descriptions and their
5       corresponding SMILES strings.
6
7       Please generate **{num_samples}** benchmark samples.
8
9       The difficulty of these samples should be: **{difficulty_description}**.
10      Examples of difficulty levels:
11      * **Beginner**: Simple acyclic molecules, common functional groups (e.g.,
12          ethanol, acetic acid, propanamine), small alkanes/alkenes/alkynes.
13      * **Intermediate**: Molecules with single or multiple rings (e.g., cyclohexane,
14          pyridine, naphthalene), basic stereochemistry (R/S, E/Z using `@@`, `/`, `\`),
15          common drugs or biomolecules (e.g., aspirin, glucose in its open-chain form).
16      * **Advanced**: Complex polycyclic systems (e.g., steroids, bridged compounds),
17          detailed stereochemistry, isotopic labeling, salts, mixtures, or reaction
18          SMILES (if the task is to represent a reaction).
19
20      For each sample, provide a JSON object with the following structure:
21
22      ```json
23      {{
24      "input": "A natural language description of a chemical compound or a task that
25          uniquely defines a chemical structure representable by a SMILES string.
26          This could be an IUPAC name, a common name, a structural description, or
27          a request to modify a base structure.",
28      "output": "The correct and valid SMILES string for the chemical structure
29          described in the 'input'. Correctness and validity are paramount."
30      }}
31      ```
32
33      **Guidelines for generating samples**:
34
35      - **Accuracy**: The generated SMILES string in the "output" field MUST
36          accurately represent the chemical structure described in the "input". Ensure
37          correct atom types, bond orders, connectivity, aromaticity, charges,
38          isotopes, and stereochemistry as implied by the input.
39      - **Validity**: All generated SMILES strings must be syntactically valid.
40      - **Clarity of Input**: The "input" description should be unambiguous and
41          provide enough information to define a specific chemical structure. Avoid
42          overly vague descriptions.
43      - **Variety**: Generate a diverse set of samples covering different chemical
44          families, structural features (rings, unsaturation, heteroatoms, functional
45          groups), and complexities according to the specified difficulty.
46
47      Output Format:
48
49      Return a JSON list containing the {num_samples} generated JSON objects.
```

Figure 15: Prompt used to generate additional samples for the SMILES task using GEMINI-2.5-PRO. Several samples were generated at the same time to increase diversity.

```
1   user
2       You are a specialized AI assistant that generates SMILES (Simplified Molecular
3       Input Line Entry System) strings from chemical descriptions. You will be given
4       a textual description of a chemical compound or a related task. Your goal is
5       to produce the most accurate and valid SMILES string representing that
6       description.
7
8       Your Task:
9
10      Based on the provided "input" description, generate the corresponding SMILES
11      string.
12
13      Output Requirements:
14
15      - Provide only the SMILES string as your output.
16      - Ensure the SMILES string is syntactically valid.
17      - Represent all specified chemical features accurately (atoms, bonds, rings,
18          aromaticity, charge, isotopes, stereochemistry).
19
20      Output:
21
22      - Provide only the smiles molecule as a raw string between triple backticks (```).
23      For instance:
24      ```smiles
25      C1=CC=CC=C1
26      ```
27
28      {sample}
```

Figure 16: Prompt used to verify samples for the SMILES task using GEMINI-2.5-PRO.

```
1  system
2      You are an expert in solving grade school math tasks. You will be presented with a
           grade-school math word problem with symbolic variables and be asked to solve it
           .\n\nBefore answering you should reason about the problem (using the <reasoning>
            field in the response described below). Intermediate symbolic expressions
           generated during reasoning should be wrapped in << >>.\n\nOnly output the
           symbolic expression wrapped in << >> that answers the question. The expression
           must use numbers as well as the variables defined in the question. You are only
           allowed to use the following operations: +, -, /, //, %, *, and **.
3
4      You will always respond in the format described below:
5      Let's think step by step. <reasoning> The final answer is <<symbolic expression>>
6
7      There are {t} trees in the {g}. {g} workers will plant trees in the {g} today. After
            they are done, there will be {tf} trees. How many trees did the {g} workers
           plant today?
8
9      Let's think step by step. Initially, there are {t} trees. After planting, there are
           {tf} trees. The number of trees planted is <<tf - t>>. The final answer is <<tf
           - t>>.
10
11     If there are {c} cars in the parking lot and {nc} more cars arrive, how many cars
           are in the parking lot?
12
13     Let`s think step by step. Initially, there are {c} cars. {nc} more cars arrive, so
           the total becomes <<c + nc>>. The final answer is <<c + nc>>.
14
15     {p1} had {ch1} {o1} and {p2} had {ch2} {o1}. If they ate {a} {o1}, how many pieces
           do they have left in total?
16
17     Let's think step by step. Initially, {p1} had {ch1} {o1}, and {p2} had {ch2} {o1},
           making a total of <<ch1 + ch2>>. After eating {a} {o1}, the remaining total is
           <<ch1 + ch2 - a>>. The final answer is <<ch1 + ch2 - a>>.
18
19     {p1} had {l1} {o1}. {p1} gave {g} {o1} to {p2}. How many {o1} does {p1} have left?
20
21     Let's think step by step. {p1} started with {l1} {o1}. After giving {g} {o1} to {p2
           }, {p1} has <<l1 - g>> {o1} left. The final answer is <<l1 - g>>.
22
23  user
24      {name} picks {n1} {fruit}s on {d1}. Then he picks {n2} {fruit}s on {d2}. On {d3}, he
            picks {mult} the number of {fruit}s he did on {d1}. How many {fruit}s does {
           name} have?
25
26  assistant
```

Figure 17: Prompt for the GSM8K-SYMBOLIC task with example user prompt in blue at the end.

```
1  user:
2      ...
3      The answer must adhere to the following grammar:
4
5      S -> Line
6      Line -> Atom OptComboChainBranchList
7      OptComboChainBranchList -> ComboChainBranchList | ε
8      ComboChainBranchList -> ComboChainBranchElement | ComboChainBranchElement
           ComboChainBranchList
9      ComboChainBranchElement -> Chain | Branch
10     Chain -> . Atom | OptBond ComboAtomRnumList
11     OptBond -> Bond | ε
12     ComboAtomRnumList -> ComboAtomRnumElement | ComboAtomRnumElement ComboAtomRnumList
13     ComboAtomRnumElement -> Atom | Rnum
14     Bond -> - | bond
15     Branch -> ( OptBondOrDotLineList )
16     OptBondOrDotLineList -> OptBondOrDotLineElement | OptBondOrDotLineElement
           OptBondOrDotLineList
17     OptBondOrDotLineElement -> OptBondOrDot Line
18     OptBondOrDot -> Bond | . | ε
19     Atom -> organicSymbol | BracketAtom
20     BracketAtom -> [ OptionalIsotope Symbol OptionalChiral OptionalHCount OptionalCharge
           OptionalMap ]
21     OptionalIsotope -> Isotope | ε
22     OptionalChiral -> chiral | ε
23     OptionalHCount -> HCount | ε
24     OptionalCharge -> Charge | ε
25     OptionalMap -> Map | ε
26     Rnum -> digit | perc digit digit
27     Isotope -> digit | digit digit | digit digit digit
28     HCount -> h digit | h
29     Charge -> + | + + | + fifteen | + digit | - | - - | - fifteen | - digit
30     Map -> : Isotope
31     Symbol -> organicSymbol | anorganicSymbol | h
```

Figure 18: For grammar prompting (G.P.), the grammar in EBNF is appended to the user prompt. The example grammar shown in blue is the SMILES grammar.

