# OpenReview forum: "Constrained Decoding of Diffusion LLMs with Context-Free Grammars"
_ICLR.cc/2026/Conference — ICLR 2026 Poster_

### Official Review · Reviewer_vXC3 · 2025-10-29

**Soundness:** 3
**Presentation:** 3
**Contribution:** 2
**Rating:** 4
**Confidence:** 5

**Summary:**

This paper introduces an approach for constraining the output of diffusion LLMs (or in general any infilling model) based on a given context free grammar. The key idea of the work is to compute the space of realizable sequences of a infilling model as a regular language and then intersect it with a target grammar to check if the set is empty. If so the proposed completion is discarded. Some optimizations are presented to make the approach practical. The evaluation shows that the approach improves the quality of pass@1 rates for several models when using constraints.

**Strengths:**

- Constraining DLM is an emerging area and an important problem and this paper gives some initial solution
- Encouraging evaluation on existing constrained decoding benchmarks

**Weaknesses:**

- The paper mostly uses well-established ideas from automata theory. E.g. I believe that the "new" algorithm for intersecting regular languages and CFGs presented in this paper is already known (https://aclanthology.org/2023.eacl-main.52/). Also the algorithm is introduced to avoid a complexity stated on line 238, but the complexity of the presented algorithm is not given (at least the reviewer could not find it). The contribution of lines 250-255 is also a bit over claiming. No-one would check emptiness of a grammar with quadratic algorithm as stated by the paper. The reviewer was quite surprised to see the authors citing a random stackexchange post for the CFG emptiness algorithm where what they describe is the algorithm given in any textbook for theory of computation (e.g., Michael Sipser, Introduction to the Theory of Computation, Thm 4.8).
- The caveats in section 3.3 are what worries me most. For Grammar Constrained Decoding, we have had many papers for which the implementations were incorrect because of how they handled the difference between tokens and lexemes. Some of these errors are reported in this paper (https://icml.cc/virtual/2025/poster/45613), which the authors should perhaps cite. The paper does not state correctness (specifically for Algorithm 3) so there is no guarantee that all and only all invalid masked sequences are rejected (something for which at the very least there should be assumptions about how the lexer operates).

**Questions:**

Is the presented algorithm sound/complete? Or when is it not? What does one need to assume about the lexer (1 lookahead?)

Is line 193 really true? If the distribution M is adversarial the algorithm will never terminate. For example, if the probability of e.g., adding an open parenthesis, is higher than ever closing it in the constrained distribution, the one will continue expanding the string without ever completing it. I don't think there is an easy way to always guarantee termination without some strong assumptions on M. This aspect is probably why the algorithm often takes many tokens and the authors need to resort to instead generating a random string in the grammar.

What does this mean: "only 7% of valid completions do not appear in the first 50 selected LLM updates?"

One of the known problems of constrained sampling is distribution distortion as discussed in this relevant related work (https://arxiv.org/abs/2405.21047). Suresh et al's Dingo have a theorem claiming that their approach to constraining DLM with regular expressions samples ("in some sense") optimally from the constrained distribution. What can the authors say about their approach?

How was the 256 tokens bound on line 336 chosen?

My understanding is that the proposed approach is not "friendly" for different sampling mechanisms (e.g., beam search) because it does not directly compute token masks? Why not directly compute token masks? The authors claim that this would require expensive preprocessing, but newer GCD algorithms do not require such expensive approaches (see llguidance and GreatGramma (https://icml.cc/virtual/2025/poster/45613), which are not cited in the paper) or at least require small-enough latency. I feel like masking would also drastically reduce the overhead as much of the overhead of the proposed method is caused by guessing and checking incorrect completions.

What do the underlined numbers mean in table 1 and 2?

For SMILES there are existing metrics for semantic quality (https://www.cs.jhu.edu/~jason/papers/lipkin+al.colm25.pdf) why using an LLM as a judge?

OTHER COMMENTS:
- I would report the absolute latency-per-token in the main body of the paper rather than the relative increase. The latter depends on the GPU on which one runs the LLM

---

> ### Author Response · Authors · 2025-11-20
> **Rebuttal (1/2)**
>
> We thank the reviewer for their insightful remarks and rigorous analysis of our approach, and provide answers to the raised questions below.
>
> ### **Please clarify the precise contributions with respect to the intersection algorithm and its complexity.**
>
> The core contribution of our work is the application of language intersection algorithms *to enable constrained decoding on diffusion language models*. The word “novel” in our work refers solely to the introduced constrained decoding algorithm.  We adjusted the wording in Section 3 to clarify our use of standard techniques for the general intersection algorithm and clarify that we do not improve its worst-case complexity.
>
> The work suggested by the reviewer [1] extends the standard intersection algorithm for DFAs with epsilon arcs, which we do not employ. Rather, our adaptations to the intersection algorithm are:
> - Choosing a suitable normal form and heuristics to maintain a small context-free grammar size.
> - Integrating the intersection algorithm and the language emptiness check into a single, efficient operation.
> - And finally, adding the required framework around the algorithm to apply it to diffusion LLMs and Multi-Region Infilling.
>
> [1] Pasti et al, *On the Intersection of Context-Free and Regular Languages*, EACL ‘23
>
> ### **Please elaborate on the soundness, completeness and distribution alignment of your algorithm, including the lexing.**
>
> We have added a section on soundness, correctness, and distribution alignment to Section 3 of the revised paper, and added more details on the overall lexing algorithm and a proof of its soundness and correctness in Appendix C. Our proof works under the assumption of maximum munch for the lexer, matching [1], which we now also cite, as recommended by the reviewer. A discussion of the alignment of our algorithm with the ideal model distribution, as proposed by [2], is presented in the answer to global question Q2.
>
> [1] Park et. al, *Flexible and Efficient Grammar-Constrained Decoding*, ICML ‘25
> [2] Park et. al, *Grammar-Aligned Decoding*, NeurIPS ‘24
>
> ### **Can you elaborate on the termination of the sampling algorithm?**
>
> The reviewer correctly points out that the algorithm would not terminate on an adversarial $M$. The corrected statement is: Since the algorithm preserves completability, there always exists a sequence of additive updates that completes $\mathbf{x}$ to be in $L$. As such, the algorithm guarantees the existence of a sequence of updates that leads to termination, but not termination itself. We have adapted the wording in Section 3 accordingly.
>
> ### **Can you elaborate on the choice of your SMILES metric in comparison to [1]?**
>
> For our evaluation dataset SMILES, we generate pairs of natural language descriptions of specific chemical molecules and their SMILES notation. In our evaluation, we provide the diffusion LM with the natural language description of the chemical molecule and task the model to write down the molecule in SMILES notation. We then compare the canonical form of the goal molecule and the generated molecule using RDKIT [2], a rule-based Python library for chemical molecule analysis.
>
> The task for the language model in [1] is to generate any drug-like compound in SMILES notation. They use another tool of RDKIT (QED) to assess whether the generated molecule has general “drug-like properties”, i.e., whether the molecule has some properties that are generally considered desirable for drugs.
>
> In particular, the metric used in [1] cannot be used to compare chemical molecules for equality, as is done in our work. Moreover, our approach uses the same chemical library as [1] to parse, canonicalize, and compare the molecules.
> We used the method of [1] to check whether our method leads to increased drug-likeness. Indeed, drug likeliness significantly increases from 0.0% in unconstrained decoding to, on average, 15.8% after our constraints are applied.
>
> [1] Lipkin et. al., *Fast Controlled Generation from Language Models with Adaptive Weighted Rejection Sampling*, COLM ‘25
> [2] https://www.rdkit.org/docs/index.html

---

> ### Author Response · Authors · 2025-11-20
> **Rebuttal (2/2)**
>
> ### **Can you elaborate on the choice of k and the maximum generation length?**
>
> We evaluated our method on a development set comprising C++, JSON, and SMILES tasks and found that universally, if the constraining leads to resampling tokens more than 50 times over the course of the generation, the generated output is correct only in 0.7% of cases. This aligns with prior work [1]: if the model can not propose even a syntactically valid token with high likelyhood, it very likely cannot generate a valid completion anymore. We therefore double this threshold to 100 to further reduce the likelihood of aborting prematurely. Further, all tested tasks can be solved within 256 tokens, which we set as a maximum output size for all methods. We added a note explaining these choices in the description of our evaluation setup in Section 4.
>
> [1] Mündler et al, *Type-Constrained Code Generation with Language Models*, PLDI ‘25
>
> ### **Please explain the highlighting in Tables 1 and 2**
> In the main tables, we use two highlights to show the significance of the results. Concretely, we compute confidence intervals of the difference between each method and each other at $95$%. We then boldface the best method and underline the methods where the confidence interval of the difference of the best method is not strictly positive.
> For example, in Table 2, the functional correctness of Dream 7B on the C++ dataset is highest with the Con. method. The difference of functional correctness of Con. to Con.${}^{-}$ is $0.7 \pm 0.7$, which includes $0$, and we thus underline the performance of Con.${}^{-}$. Meanwhile, the difference to Van. is $2.9 \pm 0.7$, which is strictly positive. Therefore, Van. is not underlined.
>
> ### **Further recommendations**
> We thank the reviewer for their further remarks and added the following changes to the paper:
>
> - We clarified that the algorithm we use in Section 3 is a standard algorithm and highlighted the importance of focusing on exploring only generating symbols.
> - We corrected the termination note on Algorithm 3 to note that the algorithm only guarantees the existence of a sequence of updates that leads to termination, but not that termination is guaranteed.
> - Section 3 now cites [1] for the language emptiness algorithm.
> - We report the absolute latency per token in the main text.
>
> [1] Sipser, *Introduction to the Theory of Computation*

---

### Official Review · Reviewer_fqDc · 2025-10-30

**Soundness:** 4
**Presentation:** 4
**Contribution:** 3
**Rating:** 8
**Confidence:** 5

**Summary:**

This paper introduces the constrained decoding method for diffusion language models, enabling them to generate text that adheres to formal grammars like context-free grammars. Unlike traditional left-to-right generation, diffusion models generate tokens in arbitrary order, which existing constrained decoding methods cannot handle. The paper solves this by formulating an "additive infilling problem" that checks whether partial outputs with holes can be completed into a valid grammar-compliant text, reducing it to testing if the intersection of the target CFG and a regular language is empty. The method achieves strong results on syntactic correctness on tasks like C++ code generation and JSON extraction while maintaining or improving functional correctness, with reasonable computational overhead.

**Strengths:**

* The paper is the first work in ensuring CFG-constrained generation with diffusion LLMs.

* The paper is well-written and easy to follow. The formalism is solid, and the problem is presented with great detail.

* The paper addressed a challenging technical problem. Additionally, there are several non-trivial technical contributions such as heuristics to reduce the size of the normalized CFG.

* The empirical results are consistently strong, showing syntactical and. Functional improvement. And I appreciate the inclusion of confidence intervals.

**Weaknesses:**

* The MRI task is not natural. Removing the arbitrary character spans is not a realistic scenario in which one would expect to use an LLM. A more realistic code will remove semantically meaningful parts of the code.

* The overhead of constraining can be large in some cases

**Questions:**

DINGO [Suresh et. al.] work ensures optimal decoding with regular grammar. How would the proposed approach compare against DINGO when using a regular grammar, both in terms of overhead and accuracy?

> All MRI models were sampled with temperature 1 and greedy decoding.

If you are using greedy decoding, shouldn’t the temp be set to 0?

---

> ### Author Response · Authors · 2025-11-20
> **Rebuttal**
>
> We thank the reviewer for their insightful remarks and provide answers to the raised questions below.
>
> ### **Can you clarify why the chosen MRI task removes random spans? How would the performance look like when it removes semantically meaningful parts?**
>
> We based our experiment on the experimental design of Bavarian et al [1], where they evaluate the performance of Fill-In-The-Middle models by asking them to fill in random spans of a golden code solution. They also suggest a similar setting, in which entire lines are removed, which are often semantically coherent and self-contained.
>
> We generate a dataset for this setting, evaluate all models on it, and add the results to Appendix D.4. Although Bavarian found random spans more difficult, we observe that models struggle more with entire lines, with the DeepSeek family of models averaging around 1% functional correctness on the task. As shown in Table 5, our constraining consistently increases syntactic and semantic correctness in this setting as well.
>
> [1] Bavarian et al. “Efficient Training of Language Models to Fill in the Middle”, arXiv.
>
> ### **How does your method compare to Grammar Prompting and the concurrent DINGO?**
>
> Please refer to our global answer to Q3, where we describe additional evaluations that compare our method to both the concurrent DINGO and Grammar Prompting as an additional baseline.
>
> ### **Please clarify whether the sampling was performed greedily or with non-zero temperature**
>
> Since we sample with non-zero temperature, we correct the statement to remove the word “greedy”, which was misplaced to differentiate with respect to other sampling methods like beam-search.

---

> > ### Comment · Reviewer_fqDc · 2025-11-26
> >
> > I thank the authors for addressing my remaining concerns. I will keep my positive score.

---

### Official Review · Reviewer_2NnJ · 2025-10-30

**Soundness:** 3
**Presentation:** 3
**Contribution:** 3
**Rating:** 6
**Confidence:** 3

**Summary:**

This paper proposes the first constrained decoding method applicable to Diffusion Language Models (DLMs), which can handle formal languages defined by Context-Free Grammars (CFGs). The method works by transforming the constrained infilling problem into determining whether the intersection between the target language (CFG) and the completion language of partial outputs (a regular language) is non-empty. Incorporating strategies such as grammar optimization and rejection sampling, it significantly improves the syntactic and functional correctness of generated results in tasks like C++ code infilling and JSON structured data extraction, while incurring only moderate computational overhead. Additionally, it also supports the Multi-Region Infilling (MRI) scenario.

**Strengths:**

- It proposes the first constrained decoding method applicable to Diffusion Language Models (DLMs), filling the gap in existing technologies that fail to constrain DLMs using Context-Free Grammars (CFGs). Meanwhile, it naturally supports the previously unsolved Multi-Region Infilling (MRI) scenario, breaking through the limitation that traditional constrained decoding can only be applied to left-to-right Prefix generation (PRE) or simple Fill-In-the-Middle (FIM).
- It achieves the first implementation of constraining models with non-fixed-order generation (DLMs) using CFGs, capable of handling complex scenarios that rely on CFG-defined syntax, such as C++, JSON, and SMILES, thereby expanding the application scope of constrained decoding.

**Weaknesses:**

- In practice, models are limited by the number of tokens, which may lead to failure in meeting syntactic constraints (such as unclosed parentheses and incomplete molecular structures) and leave some residual syntactic errors. Currently, there is a lack of efficient solutions for accurately modeling the number of remaining tokens.
- In the lexing phase, if there are a large number of ambiguous terminal sequences, even though optimization via a "unified NFA" is applied, the risk of combinatorial explosion may still arise in extreme cases, which affects inference speed.
- For tasks with strong syntactic dependence (e.g., JSON, C++), the improvement in functional correctness is significant. However, for scenarios where correct syntax does not directly determine functionality (e.g., SMILES molecular generation), only a slight improvement can be achieved (an average of 0.2%). This indicates that the method is more effective for scenarios with "strong syntax-function correlation" but has limited ability to empower tasks that require semantic understanding.

**Questions:**

In the paper, "rejection sampling" is adopted to replace the traditional masking strategy to avoid pre-inference latency. However, the relationship between the "number of sample re-sampling attempts" and "model generation diversity" during the rejection sampling process is not clearly explained. Could you please elaborate on how the threshold for the number of re-sampling attempts (such as the 100 attempts set in the paper) is determined across different tasks (e.g., C++ code generation, SMILES molecular description)? Additionally, does there exist a scenario where "excessive re-sampling attempts lead to a decline in generation diversity"?

Need to add discussion on some related work:
[1] Dong, Yixin, et al. "XGrammar: Flexible and Efficient Structured Generation Engine for Large Language Models." Eighth Conference on Machine Learning and Systems.
[2] Sun, Xintong, et al. "Earley-Driven Dynamic Pruning for Efficient Structured Decoding." Forty-second International Conference on Machine Learning.

---

> ### Author Response · Authors · 2025-11-20
> **Rebuttal**
>
> We thank the reviewer for their insightful remarks and provide answers to the raised questions below.
>
> ### **How is the resampling threshold determined (e.g., here 100) across different tasks?**
>
> We evaluated our method on a development set comprising C++, JSON, and SMILES tasks and found that universally, if the constraining leads to resampling tokens more than 50 times over the course of the generation, the generated output is correct only in 0.7% of cases. This aligns with prior work [1]: if the model does not propose even a syntactically valid token with high likelihood, it very likely cannot generate a valid completion anymore. We therefore double this threshold to 100 to further reduce the likelihood of aborting prematurely. We added a note explaining this decision in the description of our evaluation setup in Section 4.
>
> [1] Mündler et al, *Type-Constrained Code Generation with Language Models*, PLDI ‘25
>
> ### **Can you elaborate on the impact of your method on model diversity?**
>
> While we are unsure what exactly the reviewer refers to with “model generation diversity”, we provided global answers contextualizing our sampling approach (Q1) and its alignment with the original model distribution (Q2). We hope that these answers clarify the questions of the reviewer and, otherwise, kindly ask for a follow-up question.
>
> ### **Please include a mention of [1] and [2]**
>
> We have included the works in the related work and background sections where suitable.
>
> [1] Dong et al, *XGrammar: Flexible and Efficient Structured Generation Engine for Large Language Models.* MLSys ‘25
> [2] Sun et al, *Earley-Driven Dynamic Pruning for Efficient Structured Decoding*, ICML ‘25

---

### Official Review · Reviewer_9ABz · 2025-11-01

**Soundness:** 3
**Presentation:** 1
**Contribution:** 2
**Rating:** 4
**Confidence:** 4

**Summary:**

This paper introduces a constrained decoding framework for diffusion language models, which enforces structural constraints during the infilling process.

**Strengths:**

The research question that this paper studied is timely and interesting.

**Weaknesses:**

Motivation and benefit: While the paper successfully enforces syntactic validity in diffusion language models, it remains unclear whether this constraint leads to better semantic or functional outputs. Improving syntax alone doesn’t necessarily improve model accuracy or usefulness, so it would help to clarify when grammatical correctness translates to real task gains and when it simply bounds decoding behavior.

Presentation and flow: The presentation of the core algorithm (Sec. 3) feels somewhat disconnected and difficult to follow. The exposition moves rapidly from defining the infilling intersection problem to dense formal descriptions (e.g., construction of regular language, grammar intersection, normalization, and emptiness checking) without sufficient intuitive explanation or consistent narrative flow. It is not always clear how these steps tie together in the overall decoding process. Including a running example throughout this section, showing how a concrete partial program or sentence evolves through each construction would make the method more accessible.

Flexibility limitation: It seems that the proposed method constrains the model to remain within a fixed grammar throughout diffusion, which limits flexible reasoning (reasoning first then provide answer). This restriction could prevent diffusion models from leveraging intermediate free-form reasoning steps, which are often beneficial in complex tasks like code synthesis or semantic parsing.

Baselines and comparisons: The evaluation would be stronger with additional baselines such as grammar prompting [1] (provide grammar information in the prompt without enforcing it), and recent CD method on text diffusion models[2]. Reporting results across functionality, syntax validity, and runtime overhead would help contextualize the real advantages of this approach.

Efficiency and scalability: Although the paper claims practical overhead, the computational trade-offs are not deeply analyzed. A breakdown of preprocessing vs. decoding time, memory usage, and how performance scales with grammar size, number of infilling regions, or diffusion steps would strengthen the empirical section. It would also help to discuss whether the rejection-based CFG intersection introduces runtime variance or instability, especially as grammar complexity increases and validity checks become harder to perform.

[1] https://arxiv.org/abs/2305.19234
[2] https://arxiv.org/abs/2505.23061

**Questions:**

See weakness.

---

> ### Author Response · Authors · 2025-11-20
> **Rebuttal**
>
> We thank the reviewer for their insightful remarks and provide answers to their raised questions below.
>
> ### **How does output generation benefit from syntactic constraints?**
>
> Many tasks require strict adherence to syntax, for example, model tool use [1,2] or coding [3]. In these tasks, if the output does not adhere to the required syntax, it is never functionally correct. As such, syntactic constraining is frequently sought after by model users and implemented in many popular model APIs [4,5,6]. Similar to other works [2,7,8,9], we therefore focus on enabling syntactic constraining.
>
> As the reviewer points out correctly, this does not necessarily guarantee functional correctness of the generated output. However, we observe that syntactic constraints can have a significant positive impact on correctness if the model has a certain base capability, for example, in the case of C++ and JSON-Schema. Additionally, we provided concrete case studies in Appendix D.6 and a qualitative analysis of the impact of our constraints on functional improvements to show how improving syntactic constraints can also improve functional correctness.
>
> [1] Yang et al, *SWE-agent: Agent-Computer Interfaces Enable Automated Software Engineering*, NeurIPS ‘24
> [2] Wang et al, *OpenHands: An Open Platform for AI Software Developers as Generalist Agents*, ICLR ‘25
> [3] Ugare et al, *SynCode: LLM Generation with Grammar Augmentation*, TMLR ‘25
> [4] https://openai.com/index/introducing-structured-outputs-in-the-api/
> [5] https://www.claude.com/blog/structured-outputs-on-the-claude-developer-platform
> [6] https://github.com/guidance-ai/llguidance
> [7] Melcer et al, *Constrained decoding for code language models via efficient left and right quotienting of context-sensitive grammars*, arXiv
> [8] Ugare et al, *SynCode: LLM Generation with Grammar Augmentation*, TMLR ‘25
> [9] Poesia et al, *Synchromesh: Reliable code generation from pre-trained language models*, ICLR ‘22
>
>
> ### **Please revise Section 3 for additional clarity and cohesiveness.**
>
> As suggested by the reviewer, we adapt Section 3, making the connection between paragraphs clearer, adding intuitive remarks, and adding a running example. We ask the reviewer to kindly follow-up, if they find the added explanations insufficient.
>
> ### **Does your syntactic constraining allow flexible reasoning?**
>
> Yes, it is possible to combine flexible reasoning with syntactic constraining in general [1] and with our method in particular. For a concrete example of a task that first allows the model to reason in free-form and then generate an answer, please refer to the GSM8K-Symbolic evaluation we have added in Appendix D.5 of the revised paper.
>
>
> [1] Banerjee et al, *CRANE: Reasoning with constrained LLM generation*, ICML ‘25
>
> ### **How does your method compare to Grammar Prompting and the concurrent DINGO?**
>
> Please refer to our global answer to Q3, where we describe additional evaluations that compare our method to both the concurrent DINGO and Grammar Prompting as an additional baseline.
>
> ### **Please provide a detailed breakdown of the performance.**
>
> We have added a detailed analysis of computation tradeoffs in Appendix D.4 of the revised paper.
>
> As highlighted in Section 5, our approach does not require pre-computations before application. We therefore only report the runtime overhead during decoding. In Tables 3 and 4, we provide a detailed runtime analysis of our method on both MRI and DLLM evaluations, respectively. The results show a slight increase in runtime for more infilling regions (Table 3). We added a further analysis for the relationship of runtime to the size of grammars of different JSON Schemas, showing a modest growth of runtime for larger grammar sizes (Figure 6). Finally, we provide an analysis of the impact of the number of diffusion steps (Tables 5 and 6), where we show consistent syntactic and semantic improvements over all settings at similar runtime overheads.

---

### Author Response · Authors · 2025-11-20
**Global Answer to the Reviewers**

We would like to thank all reviewers for the valuable feedback and insightful questions. We are happy to see that all reviewers agree that our work tackles an important and timely research question, with some highlighting our work being the first implementation tackling this challenge, and having promising evaluation results.

We also noted common requests for more clarification and comparison to related work, which we answer below. We further answer all individually raised points in the comments to each review. Together with the answers, we have uploaded a revision of our paper and marked all changes therein in $\textcolor{purple}{\text{purple}}$ color for ease of recognition. We provide a complete list of changes at the end of this answer.

---

> ### Author Response · Authors · 2025-11-20
> **Commonly Asked Questions Q1 and Q2**
>
> ### (vXC3, 2NnJ) **Q1: Why does your method use rejection-sampling, compared to masking or hybrid approaches? Could further optimizations be applied?**
>
> Importantly, masking and rejection-sampling are equivalent in the sense that they sample a new token from the same distribution. Masking approaches are preferred when the relevant masks can be precomputed. This, however, is only possible for left-to-right generation of context-free grammars [1,2,3,4,5], and is infeasible already when requiring a specific suffix [6]. Similarly, approaches for context-sensitive languages similarly do not use masking [7,8,9]. This is because, when applying masking without being able to precompute the masks, the token mask needs to be computed during decoding, implying a feasibility check for every token in every decoding step, which is prohibitively expensive. Therefore, in such cases, rejection-sampling is the preferred approach, as it allows us to prioritize the order of feasibility checking based on the model-computed relevance of each token.
>
> It is possible to apply further optimizations to rejection-sampling and masking using hybrid approaches, for example, by using the methods proposed by [3,5]. Concretely, they pre-compute lexings of each LLM token and group them efficiently to 1) rule out tokens that are not admissible as determined by the lexer and 2) rule out classes of tokens during decoding that share similar lexing. While these approaches are likely applicable to our work, they are entirely orthogonal and introduce significant additional complexity, which is why we consider them beyond the scope of our work.
>
> [1] Ugare et al, *SynCode: LLM Generation with Grammar Augmentation*, TMLR ‘25
> [2] Poesia et al, *Synchromesh: Reliable code generation from pre-trained language models*, ICLR ‘22
> [3] Beurer-Kellner et al, *Guiding LLMs The Right Way: Fast, Non-Invasive Constrained Generation*, ICML ‘24
> [4] Dong et al, *XGrammar: Flexible and Efficient Structured Generation Engine for Large Language Models*, PMLR ‘24
> [5] Park et al, *Flexible and Efficient Grammar-Constrained Decoding*, ICML ‘25
> [6] Melcer et al, *Constrained decoding for code language models via efficient left and right quotienting of context-sensitive grammars*, arXiv
> [7] Mündler et al, *Type-Constrained Code Generation with Language Models*, PLDI ‘25
> [8] Li et al, *Correctness-Guaranteed Code Generation via Constrained Decoding*, COLM 25
> [9] Nagy et al, *ChopChop: a Programmable Framework for Semantically Constraining the Output of Language Models*, arXiv
> [10] https://github.com/guidance-ai/llguidance
>
> ### (2NnJ, vXC3) **Q2: How does the alignment of your approach with the model distribution compare to other works?**
>
> Our work uses local rejection-sampling to constrain the model distribution to the desired set of admissible outputs.  This approach is *minimally invasive*, as introduced by [1]. In particular, if the unconstrained model would generate a valid output w, then our approach generates w as well. This follows from the fact that our constraining method is complete, a proof of which we added to the revised paper in Section 3 and Appendix C.4.
>
> Further, our approach is equivalent to other common sampling or masking approaches used in the literature [1,2,3,4,5] and steers the model through local adaptations of the model distribution based on the admissibility of tokens. Therefore, it provides the estimate of the ideal output under the model distribution as outlined in [6] for “Grammar Constrained Decoding”.
>
> [6] suggest extensive exploration of future samples to determine the best aligned local choice. In principle, this approach could be used together with our constraining method to more accurately align with the model distribution. However, this comes at a significant computational cost due to the need to compute the exponentially growing number of future samples.
>
> Another option is to store all probability distributions for each next token (or group of next tokens) to be sampled and then estimate a globally optimal sampling from the stored distributions that aligns with the grammar [7]. However, this approach is currently only applicable to regular languages. An application to context-free grammars is beyond the scope of this work.
>
> [1] Beurer-Kellner et al, *Guiding LLMs The Right Way: Fast, Non-Invasive Constrained Generation*, ICML ‘24
> [2] Ugare et al, *SynCode: LLM Generation with Grammar Augmentation*, TMLR ‘25
> [3] Poesia et al, *Synchromesh: Reliable code generation from pre-trained language models*, ICLR ‘22
> [4] Dong et al, *XGrammar: Flexible and Efficient Structured Generation Engine for Large Language Models*, PMLR ‘24
> [5] Park et al, *Flexible and Efficient Grammar-Constrained Decoding*, ICML ‘25
> [6] Park et al, *Grammar-aligned decoding*, NeurIPS ‘24
> [7] Suresh et al, *DINGO: Constrained Inference for Diffusion LLMs*, arXiv

---

> ### Author Response · Authors · 2025-11-20
> **Commonly Asked Question Q3**
>
> ### (9ABz, fqDc) **Q3: Please add comparisons to related work [1, 2]**
>
> We computed the performance of Grammar Prompting [1] for our main evaluations and added them to Section 4 in the revised paper (Table 2). We find that Grammar Prompting has a mixed impact on model performance, both increasing and decreasing functional correctness, and outperforming our constrained decoding on three model-dataset combinations. This is in stark contrast to the consistent improvements that our constrained decoding provides. Importantly, the approach is orthogonal to ours, allowing users to combine them if Grammar Prompting is found beneficial.
>
> We have further included a comparison to DINGO [2] in Appendix D.1. DINGO implements constrained decoding for diffusion language models, but can only be applied to regular languages. Importantly, their implementation is not openly accessible. Therefore, we had to compare with the results reported in their work on datasets they chose. Concretely, these are a JSON dataset that is similar to our JSON evaluation and an arithmetic dataset based on GSM8K [3].
>
> We match their evaluation settings and present the numbers and further experimental details in Appendix D.5. We find that our method achieves similar performance for syntactic constraining as DINGO, and obtains the same functional correctness on the JSON dataset. On GSM8K, DINGO slightly outperforms the functional correctness of our constraining method by on average 6%. Regarding runtime overhead, we observe a similar overhead of our method compared to DINGO, with overall less than 0.3s additional latency per sample. However, our method requires no precomputation and thus markedly outperforms DINGOs' preprocessing time of up to half a minute. This is particularly critical on the JSON dataset, where preprocessing is performed for every sample.
>
>
> [1] Park et al, *Grammar-aligned decoding*, NeurIPS ‘24
> [2] Suresh et al, *DINGO: Constrained Inference for Diffusion LLMs*, arXiv
> [3] Mirzadeh et al., *GSM-Symbolic: Understanding the Limitations of Mathematical Reasoning in Large Language Models*, arXiv
> [4] de Moura and Bjørner, *Z3: an efficient SMT solver*, TACAS ‘08

---

> ### Author Response · Authors · 2025-11-20
> **List of Changes to the Paper**
>
> Here we list all major changes in our revised version of the paper. The paper PDF was updated, and all changes are highlighted in $\textcolor{purple}{\text{purple}}$ color for ease of recognition. Where entire sections or figures where changed, we highlight only the section title or figure caption.
>
> - Sec. 2.1: We added a short discussion on hybrid approaches for constrained decoding.
> - Sec. 3: We add running examples to improve clarity of the intersection algorithm.
> - Sec. 3.1.: We remove the claim that our algorithm guarantees termination.
> - Sec. 3.2: We improve connections between the paragraphs and explain which parts of our method are novel.
> - Sec. 3.4: We added a new section explaining that our method is minimally invasive, and sound and complete if the lexer follows the maximum munch principle.
> - Sec. 4.1: We clarify the choice of sampling parameters.
> - Sec. 4.3: We add a comparison with Grammar Prompting.
> - Sec. 5: We add related works suggested by the reviewers.
> - App. C.1: We add a more thorough explanation of how we handle certain edge cases in the lexing procedure.
> - Algorithm 4: We add the algorithm to complete our explanation about lexing.
> - App. C.3: We prove our method is sound and complete under certain conditions.
> - App. D.4: We add extra analysis on the runtime overhead of our method, and add an experiment where we remove entire lines from MRI instead of random subspans.
> - App. D.5: We compare our method with DINGO.
> - App. F: We add the prompts for DINGO and Grammar Prompting

---

### Author Response · Authors · 2025-12-02
**Summary of the Author-Reviewer Discussion Phase**

We thank the reviewers and AC to take the time and review our paper. We would like to provide a summary of our discussion until the closing of the discussion phase at the end of last week.

The reviewers raised two main concerns, which we addressed in the rebuttal:
First, they requested extending our comparison to related works. We provide an extensive contextualization and an experimental comparison. We show that we clearly outperform a prompting based alternative. Further, we show we have a similar performance and overhead to a concurrent method that allows only the much smaller set of regular languages as constraints.

Second, they requested more detailed statements about soundness, completeness and impact on the model distribution. We added proofs for our methods' soundness and completeness, and highlight that it is minimally invasive, as introduced in [1].

Several further requests for minor clarifications and analysis were addressed. All reported results have been integrated into the revised PDF and highlighted in $\textcolor{purple}{\text{purple}}$.
There have been no further comments or score changes from the reviewers since the posting of our comments.

[1] Beurer-Kellner et al, *Guiding LLMs The Right Way: Fast, Non-Invasive Constrained Generation*, ICML ‘24

---

### Meta-Review · Area_Chair_DL7f · 2026-01-07

**Summary:**

The paper tackles a well-defined problem using a particular approach by combining rejection sampling with formal CFGs. While this is a technically interesting and challenging problem, it was never clear if such constrained decoding approach is really appropriate in the first place and if it can even be correct while adding acceptable overhead. There are many important questions to be addressed before diving into constrained decoding in the next more complex situation. Particular to this paper, Reviewer vXC3 raised several important points that are only partly addressed by the authors. In particular,

>> Grammar Constrained Decoding, we have had many papers for which the implementations were incorrect because of how they handled the difference between tokens and lexemes. The paper does not state correctness (specifically for Algorithm 3) so there is no guarantee that all and only all invalid masked sequences are rejected (something for which at the very least there should be assumptions about how the lexer operates).

The author clarified that Algorithm 3 to note that the algorithm only guarantees the existence of a sequence of updates that leads to termination, but otherwise left the key points whether their approach is correct unaddressed.

I accept the author's response to rejection sampling which is fine for it's generality.

It's great to see the discussion and revisions to make the paper better, but still voting reject based on the detailed review of vXC3, the overall ratings, and my understanding of the contributions.

Reviewers also identified the following strengths:
* first CFG-constrained decoding for DLMs, MRI support, sound approach, non-trivial contributions
* well-written according to some reviewers
* strong validation with confidence intervals

and weaknesses:
* soundness both in the technical sense and also in the overall approach
* some results are natural consequences of automata theory, that's already well-established
* not well placed in related work

For a bit of comment outside of Meta review, how does this compare with the baseline of just trying some variations until the grammar checker is passed? Since the vanilla already has success rate in the 90%, If this was similar at an instance level, then there is no problem to begin with. What are the errors that need to be addressed anyways, and are they really defects of the model that cannot be fixed with training?

**Reviewer Concerns:**

Yes, already specified in meta review

**Reviewer Scores:**

probably not

---

### Decision · Program_Chairs · 2026-01-26

Accept (Poster)